# Validating Climate Models with Spherical Convolutional Wasserstein Distance

**Robert C. Garrett**[1]    **Trevor Harris**[2]    **Zhuo Wang**[1]    **Bo Li**[1]
[1]University of Illinois Urbana-Champaign    [2]Texas A&M University
{rcg4, zhuowang, libo}@illinois.edu
tharris@stat.tamu.edu

## Abstract

The validation of global climate models is crucial to ensure the accuracy and efficacy of model output. We introduce the spherical convolutional Wasserstein distance to more comprehensively measure differences between climate models and reanalysis data. This new similarity measure accounts for spatial variability using convolutional projections and quantifies local differences in the distribution of climate variables. We apply this method to evaluate the historical model outputs of the Coupled Model Intercomparison Project (CMIP) members by comparing them to observational and reanalysis data products. Additionally, we investigate the progression from CMIP phase 5 to phase 6 and find modest improvements in the phase 6 models regarding their ability to produce realistic climatologies.

## 1   Introduction

**Climate Model Validation**   General Circulation Models, or climate models, are mathematical representations of the climate system that describe interactions between matter and energy through the ocean, atmosphere, and land [Washington and Parkinson, 2005]. Climate models are the primary tool for investigating the response of the climate system to changes in forcing, such as increases in $CO_2$, and projecting future climate states [Flato et al., 2014]. To assess the plausibility of climate models, climate scientists compare output from model simulations against observational data [Rood, 2019]. This comparison is the focus of climate model validation techniques for ensuring that climate models capture the dynamics of the climate system [Roca et al., 2021].

The Coupled Model Intercomparison Project (CMIP) was initiated in 1995 as a comprehensive and systematic program for assessing climate models against each other and observational data [Eyring et al., 2016]. Each model in CMIP participates in a wide variety of experiments such as performing a historical simulation, a pre-industrial control simulation, and various simulations representing different scenarios for $CO_2$ emissions [Eyring et al., 2016]. Because historical simulations coincide with observational measurements, we can compare each model's synthetic climate distribution to the distribution of observational or quasi-observational data products [Raäisaänen, 2007], to assess their reconstructive skill. For complete spatial coverage we compare against reanalysis data, a blend of observations and short-range weather forecasts through data assimilation [Bengtsson et al., 2004]. This has become one popular climate model validation method [Flato et al., 2014].

**Previous approaches**   Many statistical and machine learning-based methods have been applied to assess climate model output against reanalysis fields. The most common approach is to compute the root mean square error (RMSE) between long-term means of the climate model output and the reanalysis field [Li et al., 2021, Zamani et al., 2020, Karim et al., 2020, Ayugi et al., 2021]. RMSE provides a direct measure of the differences between two climate fields but does not take into account internal variability, so we should use it with caution in evaluating climate models. Another approach,

38th Conference on Neural Information Processing Systems (NeurIPS 2024).

which is invariant to bias, is to compute measures of correlation between climate model output and reanalysis fields [Zhao et al., 2021, Zamani et al., 2020, Karim et al., 2020, Ayugi et al., 2021].

More comprehensive approaches employ techniques for random processes to compare two spatial fields. For example, Shen et al. [2002] and Cressie et al. [2008] use the wavelet decomposition to compare the spatial-frequency content of two fields. Hering and Genton [2011] measures the loss differential between models of spatial processes, Lund and Li [2009] and Li and Smerdon [2012] compare the first and second moments of two random processes, and Yun et al. [2022] identifies local differences in the mean and dependency structure between two spatiotemporal climate fields. Functional data analysis techniques have been introduced to compare spatial and spatiotemporal random fields, considering the random fields as continuous functions. Many of these approaches compare the underlying mean functions from two sets of functional data [Zhang and Chen, 2007, Horváth et al., 2013, Staicu et al., 2014]. Other approaches include the second-order structure in the comparison [Zhang and Shao, 2015, Li et al., 2016], or compare the distributions of two spatial random processes [Harris et al., 2021].

Since climate models aim to mimic the real climate which is the underlying pattern of weather, directly assessing the distributional differences between the modeled and observed data seems a more thorough approach to evaluating climate models. Vissio et al. [2020] proposed to use the Wasserstein distance (WD), a popular metric for comparing probability distributions [Villani, 2009], for such a purpose. The WD has also been considered for other use cases in climate science, such as data assimilation [Tamang et al., 2020, 2021, 2022]. The WD can be computationally expensive or even impossible to calculate between high-dimensional distributions [Kolouri et al., 2019], so Vissio et al. [2020] first converts each climate field to a single spatial mean and then only compares the distribution of spatial means. However, this dimension reduction puts their method at the risk of missing important spatial variability information, and consequently failing to accurately distinguish two climate fields that are different.

Recent contributions in the Machine Learning (ML) literature seek to compare multivariate distributions using many features while leveraging the efficiency of the one-dimensional WD [Vallender, 1974]. The sliced WD [Bonneel et al., 2015] compares random projections of distributions on $\mathbb{R}^n$, and the generalized sliced WD [Kolouri et al., 2019] extends this to a broader class of projections. The convolutional sliced WD [Nguyen and Ho, 2022] compares distributions of discrete square images in the space $\mathbb{R}^{n \times n}$, accounting for the spatial structure using kernel convolutions. However, spatial fields from climate models are defined on a spherical domain, making them incompatible with the vectorized/square nature of these distances. The spherical sliced WD [Bonet et al.] and other spherical transport methods [Quellmalz et al., 2023, Cui et al., 2019] can compare distributions of spatial point processes over a sphere, such as locations of natural disasters and extreme weather events, but cannot be used for smooth fields such as daily temperature and precipitation. Lastly, other non-WD approaches have been considered for multivariate distributional comparisons. For example, Mooers et al. [2023] compared distributions arising from global storm-resolving models using variational autoencoders.

**Our proposal**   We propose the functional sliced WD as a generalization of the sliced WD to distributions of functional data. To create a tailored tool for climate model evaluation, we define the Spherical Convolutional Wasserstein distance (SCWD) as a special case of the functional sliced WD for functions on the unit sphere $\mathbb{S}^2$, a manifold on which latitude-longitude coordinates are defined. SCWD creates slices containing a small region of spatial fields to characterize local differences in the distribution of climate variables, which are further integrated into a single measure for global differences. Compared to the spatial mean-based WD from Vissio et al. [2020], SCWD accounts for spatial features while maintaining the modest computation for univariate WD when comparing climate models against reanalysis data, resulting in a comprehensive and more robust evaluation. We apply SCWD to rank climate models and assess the progression of the new CMIP era with respect to daily average surface temperature and daily total precipitation.

## 2   Preliminaries

We consider the problem of comparing two probability distributions $P$ and $Q$. Each distribution is a member of $\mathcal{P}(\Omega)$, the set of Borel probability measures on some sample space, $\Omega$. In our application, we treat climate fields as functional data [Wang et al., 2016] over a spatial domain $\mathcal{S}$, thus we often assume $\Omega$ is a space of functions. Our sample space of interest is $L^2(\mathcal{S})$, the set of square integrable

functions from $\mathcal{S} \to \mathbb{R}$ where $\mathcal{S}$ is a compact subset of $\mathbb{R}^n$. CMIP model outputs are available at a global scale, so we consider the spatial domain to be the unit sphere $\mathbb{S}^2$, the space over which latitude and longitude coordinates are assigned to the Earth's surface. In fact, the space $L^2(\mathbb{S}^2)$ has been previously considered for modeling climate fields [Heaton et al., 2014]. To compare two functional data distributions $P$ and $Q$, we define a distance function $D(P, Q)$ to act as a similarity measure.

**Comparing Distributions of Functions**   Comparisons for distributions of functions in $L^2(\mathcal{S})$ have been studied for specific cases of $\mathcal{S}$ [Hall and Van Keilegom, 2007, Bugni and Horowitz, 2021, Pomann et al., 2016, Harris et al., 2021], but none of these have focused on $L^2(\mathbb{S}^2)$. The mathematical properties of probability measures in $\mathcal{P}(L^2(\mathcal{S}))$ have been studied before [Gijbels and Nagy, 2017, Kim, 2006], but it is challenging to calculate distances such as the WD in this space without additional assumptions [Li and Ma, 2020]. To define a similarity measure for distributions in $\mathcal{P}(L^2(\mathcal{S}))$, we build on the theory of the sliced WD and its various extensions, which have only been defined for distributions of finite-dimensional data.

**Sliced WD**   Given a Borel function $\pi : \Omega \to \mathbb{R}$, the pushforward of $P$ under $\pi$ is a valid distribution in $\mathcal{P}(\mathbb{R})$ defined as $\pi\#P(B) = P(\pi^{-1}(B))$ for all Borel sets $B$ in $\mathbb{R}$. The $r$-th order sliced WD [Bonneel et al., 2015] between $P, Q \in \mathcal{P}(\mathbb{R}^n)$ is a metric defined as the mean of the ordinary WD over univariate pushforwards:

$$SW_r(P, Q) = \left( \int_{\mathbb{S}^{n-1}} W_r(\pi_\theta \# P, \pi_\theta \# Q)^r d\theta \right)^{1/r}, \tag{1}$$

where $\pi_\theta(x) = x^T \theta$ for $x \in \mathbb{R}^n$ and $\theta \in \mathbb{S}^{n-1}$. We call $\pi_\theta$ the slicing function because it produces one-dimensional "slices" of the data using projection matrices. Because $\pi_\theta \# P$ and $\pi_\theta \# Q$ are valid distributions in $\mathcal{P}(\mathbb{R})$, the WD inside the integral can be calculated with the commonly used analytical form for univariate measures [Vallender, 1974].

**Generalized Sliced WD**   The generalized sliced WD [Kolouri et al., 2019] replaces $\pi_\theta$ with a more general class of slicing functions, denoted as $g_\theta$. Because of this increased flexibility, the generalized sliced WD is a pseudometric rather than a metric except for specific cases of $g_\theta$ [Kolouri et al., 2019]. For a given use case, the utility and metric properties of the generalized sliced WD are therefore determined by the choice of slicing function. Slicing functions can be chosen via optimization, estimated using neural networks, or specified by the researcher to isolate features of interest [Kolouri et al., 2019]. The last option is desirable for our application, allowing the slices to be restricted to spatial features of interest to climate modelers. However, the generalized sliced WD is defined between distributions of data in $\mathbb{R}^n$, a space that is not suitable for distributions of spatial fields. Climate fields could be coerced to vectors in $\mathbb{R}^n$ to be made compatible with the generalized sliced WD. However, this would result in a loss of the inherent spatial structure, making it challenging to specify a slicing function that can handle considerations such as area weighting and spatial correlations.

**Convolution Sliced WD**   The convolution sliced WD [Nguyen and Ho, 2022] represents images as matrices in the space $\mathbb{R}^{n \times n}$. The slicing function is replaced with kernel convolutions, or possibly a sequence of kernel convolutions, of $d \times d$ pixels for $d \in \mathbb{N}$. The kernel aggregates nearby locations to isolate local features, which could provide useful information to climate modelers. However, when climate fields are represented on a rectangular grid, the geographic area represented by each grid cell varies drastically between latitudes due to the non-Euclidean structure. Thus, a $d \times d$ pixel kernel will cover different sizes and shapes of geographic areas depending on location. For our application, the kernel radius should therefore be defined using geographic distance, not pixels.

## 3   Methods

We introduce the functional sliced WD framework which extends the flexible slicing process from the generalized sliced WD to the infinite-dimensional case of functions in $L^2(\mathcal{S})$. We focus on the special case of $L^2(\mathbb{S}^2)$, which we call the spherical convolutional WD (SCWD), for our climate model validation application. SCWD adapts the kernel convolution-based slicing idea of the convolutional sliced WD to a continuous setting while accounting for the non-euclidean structure of climate fields realized over the Earth's surface.

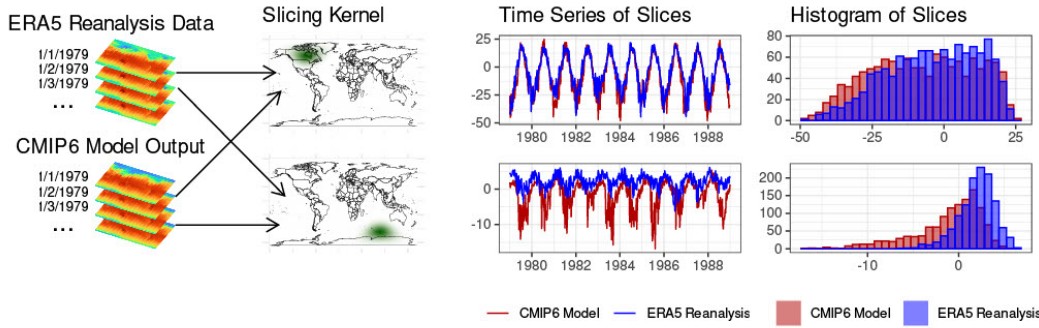

Figure 1: Diagram representing the calculation of SCWD between distributions of daily mean surface temperature (in degrees Celsius) from ERA5 and a CMIP6 model. Each day, many projections are computed using kernel convolutions, represented here at two locations. The resulting projections, called slices, summarize the local climate conditions in each dataset. The slices for each day are viewed as a sample from the marginal distribution at each location, represented here as histograms. SCWD is computed as a global mean over the univariate WD between each pair of local distributions.

## 3.1 Functional Sliced Wasserstein Distance

In general it is not possible to analytically characterize distributions in $\mathcal{P}(L^2(\mathcal{S}))$ and there are no closed form solutions for computing the WD. However, it is possible to slice elements of $L^2(\mathcal{S})$, meaning we can leverage the analytical form of the one-dimensional WD to define a computable sliced WD. We extend the convolution slicer from Nguyen and Ho [2022] to the functional data case, allowing us to project functions in $L^2(\mathcal{S})$ to values in $\mathbb{R}$ while preserving local spatial information.

**Definition 3.1** (Convolution Slicer). Let $\mathcal{S}$ be a compact subset of $\mathbb{R}^n$, $s \in \mathcal{S}$, and $k$, called the kernel function, be a continuous function from $\mathcal{S} \times \mathcal{S} \to [0, \infty)$. We define the convolution slicer $c_s(f)$, a linear operator from $f \in L^2(\mathcal{S}) \to \mathbb{R}$, as follows:

$$c_s(f) = \int_{\mathcal{S}} f(u) k(s, u) du.$$

To create a valid functional sliced WD, we construct pushforward measures based on the convolution slicer $c_s(f)$. To satisfy the definition of a pushforward measure, we must show that $c_s(f)$ is a Borel measurable function from $L^2(\mathcal{S}) \to \mathbb{R}$. By continuity of $k$, when location $s \in \mathcal{S}$ is fixed, $k(s, u)$ is a continuous function from $u \in \mathcal{S} \to \mathbb{R}$. Because $\mathcal{S}$ is compact, $k(s, u)$ is a continuous function on a compact set and is thus bounded and $L^2$-integrable. It follows that the convolution slicer $c_s(f)$ is an integral of the product of two functions $f, k \in L^2(\mathcal{S})$, so by Hölder's inequality, $c_s(f)$ is a bounded linear operator from $L^2(\mathcal{S}) \to \mathbb{R}$. Stein and Shakarchi [2011] states that bounded linear operators are also continuous, so $c_s(f)$ is a continuous linear operator and thus Borel measurable. So, for any measure $P \in \mathcal{P}(L^2(\mathcal{S}))$, the pushforward $c_s \# P$ is a valid measure in $\mathcal{P}(\mathbb{R})$. Therefore, we can define a functional sliced WD between distributions in $\mathcal{P}(L^2(\mathcal{S}))$ as follows:

**Definition 3.2** (Functional Sliced WD). Let $\mathcal{S}$ be a compact subset of $\mathbb{R}^n$, $r \geq 1$, and $P, Q \in \mathcal{P}(L^2(\mathcal{S}))$. Let $c_s$ be an operator satisfying Definition 3.1. We define the ($r$-th order) functional sliced WD between $P$ and $Q$ as follows:

$$FSW_r(P, Q) = \left( \int_{\mathcal{S}} W_r(c_s \# P, c_s \# Q)^r ds \right)^{1/r}$$

where $W_r$ is the Wasserstein metric on $\mathcal{P}(\mathbb{R})$.

Because $c_s \# P$ and $c_s \# Q$ are valid univariate probability measures, the analytical form of the univariate WD can be applied for efficient calculations. We introduce theoretical properties for the functional sliced WD in Theorem 3.3

**Theorem 3.3.** *For all compact subsets $\mathcal{S} \subset \mathbb{R}^n$, $FSW_r$ is a pseudometric on $\mathcal{P}(\mathcal{F}_{\mathcal{S}})$ and maintains the $r$-convexity property of the ordinary $W_r$ metric.*

Proof of Theorem 3.3 is provided in Appendix A. It is unknown if the final positivity property of a metric is satisfied. Proof of this property would require an invertible Radon-like transformation to be defined for probability measures in $\mathcal{P}(L^2(\mathcal{S}))$. Therefore, as with the generalized sliced WD, it is up to the researcher to specify an appropriate kernel function over the domain of interest. We provide such a choice for our application and give theoretical justification in Section 3.2.

## 3.2 Spherical Convolutional Wasserstein Distance

For our application to climate model validation, we specify the spatial domain $\mathcal{S}$ to be the unit sphere $\mathbb{S}^2$, the space over which latitude-longitude coordinates are assigned to locations on the Earth. To preserve local spatial information, we specify the slicing function to be a radial kernel function. We introduce the spherical convolutional WD (SCWD) as a specific case of the functional sliced WD:

**Definition 3.4** (Spherical Convolutional WD). Let $P, Q \in \mathcal{P}(L^2(\mathbb{S}^2))$ and $r \geq 1$. We define the ($r$-th order) SCWD between $P$ and $Q$ as follows

$$SCW_r(P,Q) = \left( \int_{\mathbb{S}^2} W_r(\omega_s \# P, \omega_s \# Q)^r ds \right)^{1/r}, \qquad \omega_s(f) = \int_{\mathbb{S}^2} f(u)\phi(s,u)du,$$

where $\omega_s$ is a convolution slicer that satisfies Definition 3.1 with associated radial kernel $\phi(s, u)$. Because $\phi$ is a radial kernel function, $\omega_s$ aggregates local information and the resulting pushforward measures $\omega_s \# P$ and $\omega_s \# Q$ represent the local distribution around each location $s$. SCWD is therefore calculated as the global mean of the WD between local distributions at each location. The local WD values can be recorded and later visualized as a map to pinpoint regions with higher or lower similarity. Figure 1 demonstrates the process for calculating SCWD between two distributions of surface temperature fields, and details on the implementation are provided in Appendix B.

**Kernel** In our analysis, we specify the kernel function to be $\phi(|s - u|; l)$, where $|s - u|$ is the chordal distance between $s$ and $u$ and $\phi$ is the Wendland kernel function used in Nychka et al. [2015]:

$$\phi(d; l) = \begin{cases} (1-d)^6(35d^2 + 18d + 3)/3 & d \leq l, \\ 0 & d > l, \end{cases} \tag{2}$$

with range parameter $l > 0$ determining the radius over which the kernel is nonzero. The Wendland kernel meets the continuity assumption in Definition 3.1 and is also compact, which enables efficient sparse computations for our analysis.

Positive definite kernels, such as the Wendland kernel for $l$ less than the diameter [Hubbert and Jäger, 2023], allow us to retain full spatial information via the spectral density. This is because the convolution theorem on $\mathbb{S}^2$ [Driscoll and Healy, 1994] gives an injective correspondence between the spectral density of a function $f \in L^2(\mathbb{S}^2)$ and the spectral density of the convolution $(f * k)(s) = \int_{\mathbb{S}^2} f(u)k(s,u)du$ when $k$ is positive definite. Note that as $l \to \infty$, the Wendland kernel converges to the flat kernel $\phi(d; \infty) = 1$, resulting in a SCWD where every slice is the global mean. In this case, the SCWD will be equal to the global mean-based WD from Vissio et al. [2020], leading to a complete loss of spatial variability information. In our analysis, we ensure a positive definite kernel by specifying $l$ to be less than the diameter of the Earth (about $12,750$ km). We study the sensitivity of our results to this parameter in Section 4.4.

**Spatial Analysis** Projection selection approaches such as the Max-Sliced WD [Deshpande et al., 2019] and Energy-Based Sliced WD [Nguyen and Ho, 2024] have been introduced as alternatives to the sliced WD that give higher slicing weight to the directions of greatest variability between high-dimensional distributions. Each of our slices corresponds to a local mean around a specific location, so applying projection selection methods to SCWD would be equivalent to identifying the geographic regions in which two distributions of spatial fields have the greatest differences. This is of keen interest when validating climate models, so we examine spatial maps of our slices in Section 4.2 to identify these regions. However, for a comprehensive evaluation of climate fields, we favor the geographically-balanced SCWD in Definition 3.4 when evaluating similarity to reference datasets.

## 4 Climate Model Validation

We consider climate model outputs from the Coupled Model Intercomparison Project (CMIP) historical experiment phases 5 and 6. We focus on daily average near-surface (2m) temperature in degrees

Celsius and daily total precipitation in mm. The CMIP6 historical simulations are organized by ensembles, each of which is distinguished with an `ripf` identifier (`rip` for CMIP5), representing realization, initialization, physics, and forcings of the model, respectively [Eyring et al., 2016]. We obtain 46 CMIP6 model outputs with the `r1i1p1f1` ID and 33 CMIP5 model outputs with the `r1i1p1` ID. Output was obtained either at the daily frequency or aggregated from 3-hourly data. At the time of writing, two of the 79 total models did not have output available at a suitable frequency for surface temperature. All 79 models had output available for total precipitation.

To serve as references for climate model evaluation, we collect the European Centre for Medium-Range Weather Forecasts (ECMWF) Reanalysis 5th Generation (ERA5) [Hersbach et al., 2020] as well as the Reanalysis-2 data from the National Centers for Environmental Protection (NCEP) [Kanamitsu et al., 2002] reanalysis datasets. Both datasets were obtained for surface temperature and total precipitation at a daily frequency. Due to known issues with reanalysis data for precipitation [Tapiador et al., 2017], we obtain observations from the National Centers for Environmental Information (NCEI) Global Precipitation Climatology Project (GPCP) Daily Precipitation Analysis Climate Data Record [Huffman et al., 2001, Adler et al., 2020] as an additional reference for precipitation.

The historical time periods for each climate variable were chosen to maximize the available model outputs and reference datasets. For surface temperature, a common time period of January 1, 1979 to November 30th, 2005 was collected for each CMIP output and reanalysis dataset. For total precipitation, we restrict the time period to October 1, 1996 to November 30th, 2005 to accommodate the first available day of observations in the GPCP dataset. Each data product represents the climate variables on a different latitude-longitude grid, which varies in size and structure. See Appendix C for full details on the spatial resolution and availability of temperature/precipitation for each dataset.

## 4.1 Evaluation of CMIP6 Models

To evaluate the skill of CMIP6 models in characterizing historical climate distributions, we compute SCWD between each model output and the reference datasets. Models which excel at replicating the local climate distribution in many different areas of the Earth will have a low SCWD. Conversely, models which fail to capture features of the local climate distributions, such as the mean, variance, or extremes, will have a higher SCWD. We do not expect perfect agreement between models and historical data, so, similar to Vissio et al. [2020], we additionally calculate SCWD between the reference datasets as a baseline for comparison. All SCWD calculations in this section use the Wendland kernel with range parameter of 1,000km for slicing.

We select the ERA5 Reanalysis as our reference dataset for (2m) surface temperature due to its high spatial resolution. We calculate distances from each surface temperature model output in our CMIP5 and CMIP6 ensembles to ERA5. In addition, we calculate SCWD between ERA5 and the NCEP Reanalysis to compare the variability between reanalysis datasets to the variability between models and reanalysis. For total precipitation, we select the GPCP observational data as our reference. We calculate distances from each total precipitation model output in our CMIP5 and CMIP6 ensembles to GPCP. We include SCWD calculations from GPCP to both ERA5 and NCEP for comparison, with the secondary goal of assessing the accuracy of each reanalysis in faithfully filling gaps in observed precipitation measurements. Full details on the SCWD rankings can be seen in Tables 2, 3, 4, and 5 in Appendix D. Here, we focus on the results for CMIP6, and Figure 2 provides the SCWD from each CMIP6 model output to the ERA5 surface temperature field and GPCP total precipitation field.

**Surface Temperature**    For surface temperature, NCEP has a lower SCWD to ERA5 than all CMIP models. This is not a surprise: both ERA5 and NCEP are based on observations, so we expect their temperature distributions to be similar in most regions. Among the model outputs, many models have a SCWD to ERA5 similar to that of NCEP. In particular, AWI-CM-1-1-MR from the Alfred Wegener Institute and MPI-ESM1-2-HR from the Max Planck Institute have the lowest SCWD for surface temperature, with a few models close behind.

**Total Precipitation**    Compared to surface temperature, NCEP no longer has a lower SCWD to GPCP than the CMIP6 models. Instead, the ERA5 total precipitation field, which has the lowest SCWD to GPCP, serves as a better baseline for comparison. Deficiencies of precipitation from reanalysis have been reported in previous studies [Janowiak et al., 1998]. In brevity, precipitation is sensitive to model physics and is not strongly constrained by observations via data assimilation. The low SCWD of ERA5 can likely be attributed to the high model resolution and more advanced model physics and data assimilation system compared to NCEP. Among the CMIP6 models, the

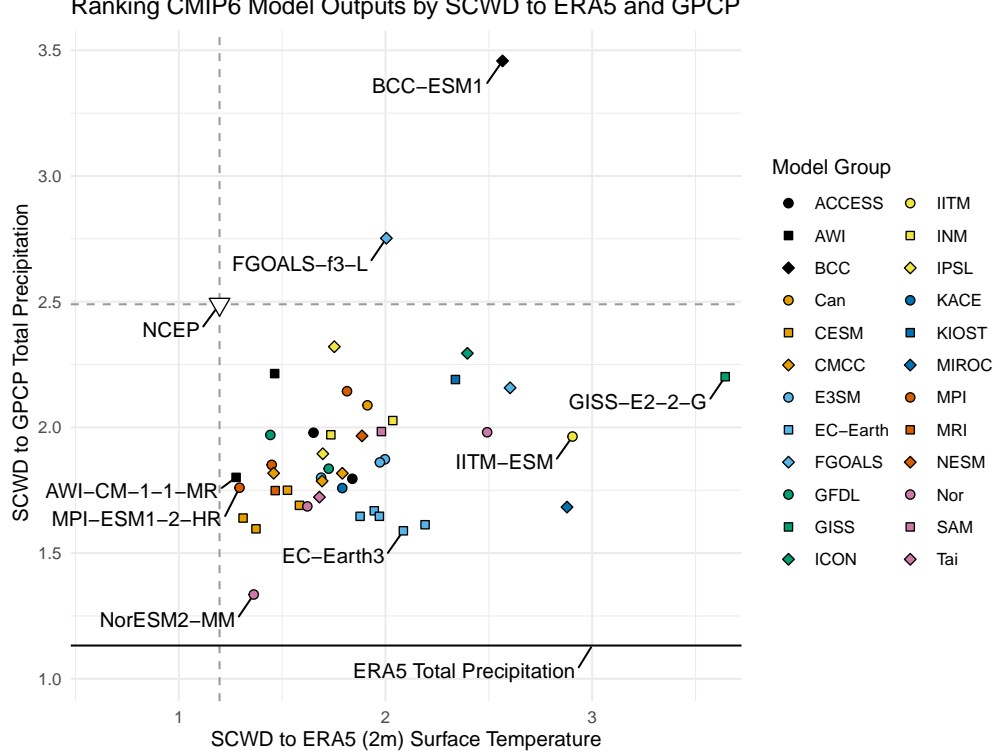

Figure 2: Ranking CMIP6 model outputs using SCWD. Each model output is represented by a point on the scatter plot and models from the same group share the color and shape. The x-axis and y-axis values represent each model's SCWD to the ERA5 surface temperature and GPCP total precipitation fields, respectively. The NCEP reanalysis is included as a blank triangle with dashed lines representing the SCWD to ERA5 and GPCP. The SCWD from the ERA5 total precipitation field to GPCP is represented as a solid line.

Norwegian Climate Centre NorESM2-MM model output stands out with the lowest SCWD to GPCP by a relatively wide margin. The EC-Earth3 and CESM model outputs also form clusters with low SCWD values for precipitation.

No single model has the lowest SCWD value for both surface temperature and total precipitation, but NorESM2-MM seems to have the best balance of low distances for each variable and is not far off from the intersection of the lines for our reference datasets. Similarly, no model has the highest SCWD value for both climate variables. The GISS-E2-2-G model from NASA's Goddard Institute for Space Studies is a distinct outlier with a high surface temperature SCWD to ERA5, and the Beijing Climate Center BCC-ESM1 model is an outlier in terms of high SCWD to GPCP. We investigate these high SCWD values in Section 4.2.

## 4.2 Spatial Comparisons

Because SCWD is calculated as a global mean of local WD values, we can investigate the geographic sources of these outlying high SCWD values. Figure 3 provides a spatial breakdown of the local WD values obtained when calculating SCWD for surface temperature between ERA5 and AWI-CM-1-1-MR as well as ERA5 and GISS-E2-2-G, the models with the lowest and highest SCWD to ERA5, respectively. Overall, both maps seem smooth or continuous in space, with little variation between neighboring locations in most cases. For the map between AWI-CM-1-1-MR and ERA5, the local WD values are relatively low everywhere, with regions of slightly higher values near the poles and mountains. Compared to AWI-CM-1-1-MR, the GISS-E2-2-G model has similarly low WD values in the tropics. However, closer to the poles, the local WD values begin to increase. In particular, the Arctic region has extremely high WD values relative to the rest of the Earth. This is indicative of the previously documented winter cool bias in the Artic region for GISS-E2-2-G [Kelley et al., 2020].

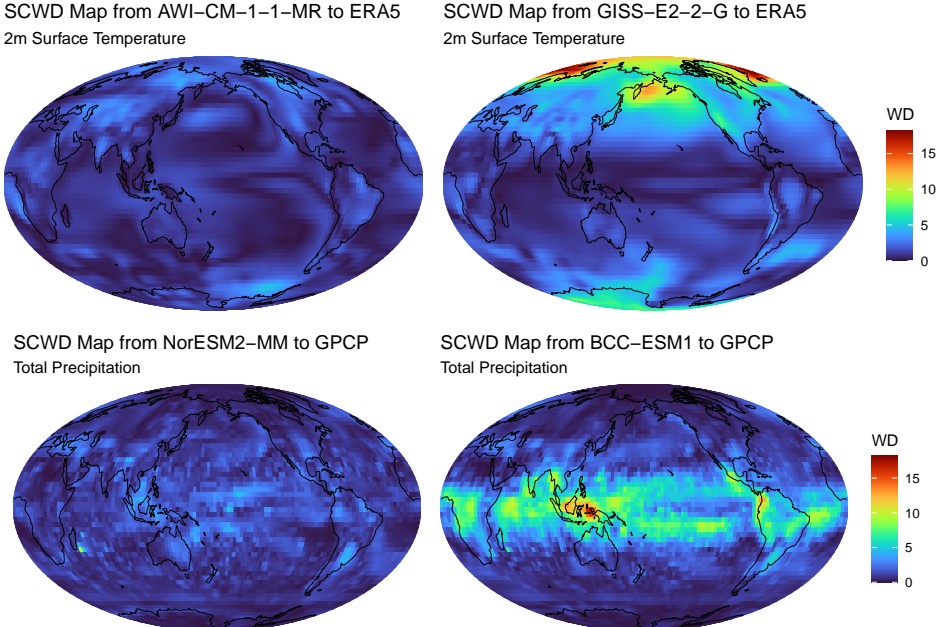

Figure 3: Top: Map of local Wasserstein distances from ERA5 to two CMIP6 2m surface temperature outputs: AWI-CM-1-1-MR and GISS-E2-2-G. Bottom: Map of local Wasserstein distances from GPCP to two CMIP6 total precipitation outputs: NorESM2-MM and BCC-ESM1. Color fill at each location is determined by the WD between the local distributions obtained from the convolution slicer in Definition 3.4. The color scale is shared for all maps and continental boundaries are included in black to aid spatial comparisons.

Similar maps are provided for the NorESM2-MM and BCC-ESM1 outputs compared to GPCP total precipitation. Overall, both maps are less smooth than the surface temperature maps, potentially due to the more localized nature of precipitation. Looking at both models, the higher values of SCWD near the equator in the Pacific and Atlantic oceans may be related to the double-Intertropical Convergence Zone problem common in CMIP models, in which excessive precipitation is produced in the southern tropics [Mechoso et al., 1995]. This trend is much more pronounced for BCC-ESM1 than for NorESM2-MM. Additionally, BCC-ESM1 has a region of particularly high WD values around eastern Indonesia, where wind-terrain interaction plays an important role in the regional distribution of precipitation. This region has been previously highlighted in Zhang et al. [2021] as an area where BCC-ESM1 heavily overestimates annual mean precipitation. SCWD maps for both climate variables and all CMIP5/CMIP6 model outputs are provided in the supplemental material.

## 4.3 Comparing CMIP5 and CMIP6

To assess the progression from CMIP5 to CMIP6, Figure 4 provides boxplots of the SCWD from each CMIP model to the reference datasets. The left panel provides SCWD calculations from the surface temperature models and NCEP to the ERA5 Reanalysis. The median SCWD value for CMIP6 models to ERA5 is lower than that of CMIP5. However, the two boxplots share a similar range, so the difference between the CMIP5 and CMIP6 ensembles is subtle. Overall, we see a promising, albeit limited, decrease in SCWD for typical CMIP6 models compared to CMIP5. This indicates improved performance of CMIP6 when it comes to reconstructing realistic temperature distributions at the local level. The right panel provides SCWD calculations from each precipitation model and ERA5/NCEP to the GPCP dataset. The median SCWD value for CMIP6 is again lower than that of CMIP5. Compared to surface temperature, the difference between CMIP5 and CMIP6 is more distinct. This indicates that relative to surface temperature, precipitation modeling has seen greater gains with the transition from CMIP5 to CMIP6. The improvement of CMIP6 models compared to CMIP5 in precipitation representation has been reported in previous studies using different evaluation

methods (e.g., Chen et al. [2021]). It can be probably attributed to the more advanced model physics and overall higher model resolution in CMIP6.

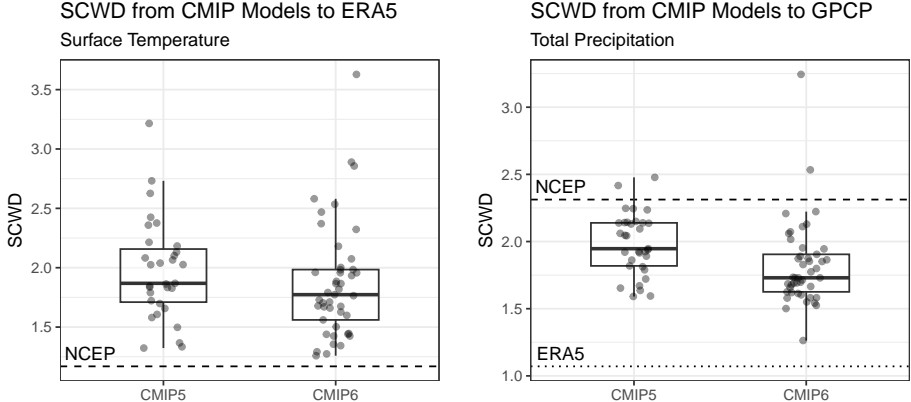

Figure 4: Left: Boxplots of SCWD from the CMIP5 and CMIP6 model outputs to the ERA5 Reanalysis for 2m surface temperature. Right: Boxplots of SCWD from the CMIP5 and CMIP6 model outputs to the GPCP observational dataset for total precipitation. Each plot contains points representing the SCWD from each CMIP model output to the reference dataset (ERA5 or GPCP), with CMIP5 and CMIP6 separated into two boxplots for comparison. Dotted and dashed lines are included to represent the SCWD from NCEP to ERA5/GPCP and ERA5 to GPCP, respectively.

## 4.4 Metric Comparison

We compare our method to baseline methods and evaluate the sensitivity of our results to the choice of range parameter. We generate seven additional rankings for the CMIP models: SCWD with a range of 500km and 2,500km respectively, the global mean-based WD (GMWD) from Vissio et al. [2020], RMSE and MAE computed on long-term mean climatologies, and WD and Sliced WD on data regridded to an icosahedral grid to preserve area weighting. Technical details and rankings together with our original 1,000km SCWD results can be seen in Tables 2 and 3 in Appendix D.

All SCWD rankings are nearly identical regardless of the range parameter, except that several precipitation models, such as GFDL-CM4 and E3SM-2-0, rank better with a larger range. From here forward, we focus only on the chosen 1,000km range parameter for SCWD. For most baseline methods (RMSE, MAE, WD, and Sliced WD), their rankings are moderately similar to those of SCWD, indicating general agreements, though each metric still shows unique perspectives. However, there is a large discrepancy between the rankings of GMWD and all other rankings. For surface temperature, SCWD, RMSE, and WD all show that the distance from NCEP to ERA5 is lower than that of all CMIP6 models, and MAE and Sliced WD also rank NCEP favorably among the models. However, GMWD ranks NCEP only among the median CMIP6 models. For precipitation, we see a similar result for ERA5 compared to GPCP: ERA5 has the best ranking when using SCWD, RMSE, MAE, and WD, but is third in the rankings for Sliced WD and again is around the median for GMWD.

To better understand the differences in rankings between SCWD and baseline model evaluation methods such as GMWD and RMSE/MAE, Appendix E provides an experiment and data example. The experiment shows a case where SCWD detects changes in both the climatological mean and the variance of the anomalies, while the other methods detect only one type of change. In particular, we show that the RMSE and MAE criteria are unable to detect isolated changes in the variance of the anomalies. The data example focuses on the SAM0-UNICON surface temperature model. Despite the overall similarity in rankings for CMIP6 surface temperature, this model ranks significantly higher by RMSE/MAE than by SCWD. Further investigation shows that SAM0-UNICON exhibits significant differences in both the climatological mean and the variance of the anomalies. Because RMSE/MAE are unable to detect shifts in the variance of the anomalies they artificially inflate the ranking of this model compared to SCWD.

# 5 Discussion

Climate model validation is critical for ensuring that climate models faithfully represent the Earth system. To this end, we developed a new similarity measure, called spherical convolutional Wasserstein distance (SCWD), which quantifies model performance in a way that properly accounts for spatial variability. SCWD builds on previous sliced WD methods to compare distributions of infinite-dimensional functional data, specifically surfaces on the sphere $\mathbb{S}^2$. Overall, our results indicate that incorporating local perspectives, rather than just the global mean, is essential when evaluating the similarities between climate models and observational data. In both theory and practice, a model that represents the global mean well may have large compensating errors at the regional scale. SCWD better represents the regional performance of climate models than the previously proposed WD-based evaluation criteria in [Vissio et al., 2020]. Furthermore, SCWD is not limited to only comparing the long-term mean climate state as with RMSE and MAE, and our convolution slicing provides better spatial insights (in the form of maps) than WD and Sliced WD.

For surface temperature, we found evidence to suggest that SCWD is more accurate than the previous WD-based approach for evaluating climate models, such as the two reanalysis datasets being more similar compared to climate models and reanalysis. However, for precipitation, similar findings were made for only one of the reanalysis datasets. Given the limitations with reanalysis for precipitation discussed in Section 4.1, additional experiments from different perspectives may be needed to fully assess the accuracy of our method. Because we have only shown that SCWD is a pseudometric, rather than a metric, SCWD may not be able to fully distinguish between climate fields which have differences only in the joint spatial distribution, rather than at the local level. We acknowledge that determining the sample complexity of SCWD is an area that requires further investigation.

We hope our method will be tested under many different scenarios and variables in climate science. For example, calculating SCWD on monthly mean or even annual data could help to detect biases in terms of longer range temporal variability. Another idea is to remove long-term climatological means from the data before computing SCWD. This would provide additional insight into which features are being captured beyond the mean climate state. SCWD can also be applied to machine-assisted climate model tuning [Hourdin et al., 2017], a growing area of research that relies on quality model evaluation metrics. Some ideas for future work to further improve SCWD include learning an optimal range value for the Wendland kernel function or using a neural network to estimate the kernel function, similar to Kolouri et al. [2019].

Outside of climate science, SCWD can be used as a loss function for training generative models for $360^o$ images. This only requires a straightforward extension of SCWD to allow for multiple convolution layers, similar to the rectangular case in Nguyen and Ho [2022]. Likewise, the more general functional sliced WD can be used as a loss function for generative models on any functional manifold. For example, we can apply the functional sliced WD to texture mapping or color transfer on the surface of 3D models, which are typically considered as non-euclidean manifolds. SCWD can be also adapted to compare distributions of spatiotemporal fields, rather than spatial fields as in this article, by including both time and space in the functional data domain. This would require specifying a space-time kernel function [Porcu et al., 2016]. The more functional sliced WD framework has a variety of potential use cases for comparing distributions of functions on a broad class of manifolds. Potential application areas where the data lie on non-trivial manifolds include facial recognition [Li et al., 2014], astronomy [Szapudi, 2008], and ecology [Sutherland et al., 2015].

## Acknowledgments and Disclosure of Funding

We acknowledge the World Climate Research Programme, which, through its Working Group on Coupled Modelling, coordinated and promoted CMIP6. We thank the climate modeling groups for producing and making available their model output, the Earth System Grid Federation (ESGF) for archiving the data and providing access, and the multiple funding agencies who support CMIP6 and ESGF. NCEP/DOE Reanalysis II data provided by the NOAA PSL, Boulder, Colorado, USA, from their website at `https://psl.noaa.gov`. This work is partially supported by the National Science Foundation grants NSF-DMS-1830312, NSF-DGE-1922758, and NSF-DMS-2124576 as well as the National Oceanic and Atmospheric Administration (NOAA) grant NA18OAR4310271. Lastly, we thank the referees and area chairs for their time and valuable feedback.

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

# A  Proof of Theorem 3.3

*Proof.* Let $r \geq 1$, let $\mathcal{S}$ be a compact subset of $\mathbb{R}^n$, and let $P, Q, U \in \mathcal{P}(L^2(\mathcal{S}))$. We show the identity property holds:

$$FSW_r(P, P) = \left( \int_{\mathcal{S}} W_r(c_s \# P, c_s \# P)^r ds \right)^{1/r}$$
$$= \left( \int_{\mathcal{S}} 0^r ds \right)^{1/r}$$
$$= 0$$

The second line holds by the identity property of the ordinary WD. Next, the symmetry property:

$$FSW_r(P, Q) = \left( \int_{\mathcal{S}} W_r(c_s \# P, c_s \# Q)^r ds \right)^{1/r}$$
$$= \left( \int_{\mathcal{S}} W_r(c_s \# Q, c_s \# P)^r ds \right)^{1/r}$$
$$= FSW_r(Q, P)$$

The second line holds by the symmetry property of the ordinary WD. Next, the triangle inequality:

$$FSW_r(P, U) = \left( \int_{\mathcal{S}} W_r(c_s \# P, c_s \# U)^r ds \right)^{1/r}$$
$$\leq \left( \int_{\mathcal{S}} [W_r(c_s \# P, c_s \# Q) + W_r(c_s \# Q, c_s \# U)]^r ds \right)^{1/r}$$
$$\leq \left( \int_{\mathcal{S}} W_r(c_s \# P, c_s \# Q)^r ds \right)^{1/r} + \left( \int_{\mathcal{S}} W_r(c_s \# Q, c_s \# U)^r ds \right)^{1/r}$$
$$= FSW_r(P, Q) + FSW_r(Q, U)$$

The second line holds by the triangle inequality property of the ordinary WD and the third line holds by the Minkowski inequality. Lastly, we check the $r$-convexity property. Let $\lambda \in [0, 1]$. If we take $P$ as the reference distribution, this property tells us what happens to the functional sliced WD as we interpolate between $Q$ and $U$:

$$FSW_r(P, \lambda Q + (1 - \lambda)U) = \left( \int_{\mathcal{S}} W_r(c_s \# P, c_s \# [\lambda Q + (1 - \lambda)U])^r ds \right)^{1/r}$$
$$= \left( \int_{\mathcal{S}} W_r(c_s \# P, \lambda c_s \# Q + (1 - \lambda)c_s \# U)^r ds \right)^{1/r}$$
$$\leq \left( \int_{\mathcal{S}} \lambda W_r(c_s \# P, c_s \# Q)^r + (1 - \lambda)W_r(c_s \# P, c_s \# U)^r ds \right)^{1/r}$$
$$\leq \left( \lambda \int_{\mathcal{S}} W_r(c_s \# P, \lambda c_s \# Q)^r ds \right)^{1/r} +$$
$$\left( (1 - \lambda) \int_{\mathcal{S}} W_r(c_s \# P, c_s \# U)^r ds \right)^{1/r}$$
$$= \lambda^{1/r} FSW_r(P, Q) + (1 - \lambda)^{1/r} FSW_r(P, U)$$

The second line follows from properties of linear operators (in this case $c_s$). The third line follows by the $r$-convexity of the ordinary WD, and the fourth line follows from the Minkowski inequality. We have shown all three pseuduometric properties and the $r-$convexity property, so our proof is complete. $\qquad\square$

# B  SCWD Implementation

The following algorithm describes the process by which we calculated the SCWD values shown in Section 4. All computations were performed using R version 4.3.2 on an Ubuntu operating system with an Intel i5-9600k processor (6 cores), 32GB RAM, and 3TB hard disk space. Total computation time for all experiments was under 72 hours. The calculation of each distance measure was fast, but loading the required datasets into memory for processing took the majority of the compute time.

---

**Algorithm 1** Spherical Convolutional Wasserstein Distance Approximation

DATA
   Reference dataset $X_0(t), t \in \mathcal{T}_0$: sample of spatial fields
   Model outputs $X_1(t), t \in \mathcal{T}_1, ..., X_n(t), t \in \mathcal{T}_n$: $n$ samples of spatial fields to be compared to $X_0$

PARAMETERS AND APPROXIMATION GRIDS
   Wasserstein order parameter $r$: 2
   Range parameter $l$: 1,000 km kernel radius
   Approximation quantiles $Q$: 200 evenly spaced quantiles ranging from 0 to 1 with a step size of 0.005
   Grid $G_1$: $60 \times 120$ regular latitude-longitude grid of center points for strided convolution
   Grid $G_2$: $361 \times 720$ regular latitude-longitude grid to provide discrete approximation of spherical domain

STEP 1. PRECOMPUTE SLICING WEIGHTS
   **for** each location $s \in G_1$ **do**
      Calculate vector of chordal distances from $s$ to all locations in $G_2$
      Calculate Wendland function (2) with range $l$ for all locations in $G_2$ using distance vector
      Apply area weighting to the Wendland kernel values using the area of each grid cell in $G_2$
      Normalize area-weighted kernel and store the results as a sparse vector $W(s)$
   **end for**

STEP 2. COMPUTE SLICED QUANTILES
   **for** $i \in 0, 1, ..., n$ **do**
      Re-grid $X_i$ to $G_2$ without smoothing (one nearest neighbor upsampling)
      **for** each location $s \in G_1$ **do**
         Slice $X_i$ into one dimension using the dot product $X_i^*(s,t) = \langle W(s), X_i(t) \rangle$
         Calculate the sliced quantile function of $X_i^*(s,t)$, denoted as $F_i^{-1}(s,q), q \in Q$
      **end for**
   **end for**

STEP 3. CALCULATE APPROXIMATE SCWD
   **for** $i \in 1, ..., n$ **do**
      **for** each location $s \in G_1$ **do**
         Calculate the local WD between $X_0$ and $X_i$ centered around $s$ as $d_i(s)^r = \sum_{q \in Q} |F_0^{-1}(q) - F_i^{-1}(q)|^r$
      **end for**
      Calculate the approximate SCWD between $X_0$ and $X_i$ as $SCWD(X_0, X_i) \approx \left( \sum_{s \in G_1} d_i(s)^r \right)^{1/r}$
   **end for**

---

The climate model outputs and reanalysis datasets (ERA, NCEP) used in our analysis have no missing data. However, the GPCP observational dataset used as the reference for total precipitation did have missing data at some sites for a few days in the historical period. To handle missing data in the GPCP dataset, we modified the slicing process in Step 2. If locations corresponding to greater than 50% of the convolution weight were missing when calculating a slice value, an NA value was recorded. The local WD calculations in Step 3 were computed ignoring these NA values. In total, 17,943 slices were missing sufficient data when using the 50% threshold. However, a total of 24,105,600 slices were considered for the GPCP dataset (3,348 days in the analysis period with 7,200 slices per day in the strided convolution), so the missing slices constituted under 0.1% of the final sliced GPCP data used in the analysis.

# C    Table of Data Details and Access Links

| Obs./Reanalysis Data | Longs | Lats | TAS | PR |
|---|---|---|---|---|
| NCEP Reanalysis | 144 | 73 | Yes | Yes |
| ERA5 Reanalysis | 1440 | 721 | Yes | Yes |
| GPCP Observations | 360 | 180 | No | Yes |

| CMIP5 Models | Longs | Lats | TAS | PR |
|---|---|---|---|---|
| ACCESS1-0 | 192 | 145 | Yes | Yes |
| ACCESS1-3 | 192 | 145 | Yes | Yes |
| CanCM4 | 128 | 64 | No | Yes |
| CMCC-CESM | 96 | 48 | Yes | Yes |
| CMCC-CM | 480 | 240 | Yes | Yes |
| CMCC-CMS | 192 | 96 | Yes | Yes |
| CNRM-CM5 | 256 | 128 | Yes | Yes |
| CSIRO-Mk3-6-0 | 192 | 96 | Yes | Yes |
| CanESM2 | 128 | 64 | Yes | Yes |
| EC-EARTH | 320 | 160 | Yes | Yes |
| FGOALS-g2 | 128 | 60 | Yes | Yes |
| FGOALS-s2 | 128 | 108 | Yes | Yes |
| GFDL-CM3 | 144 | 90 | Yes | Yes |
| GFDL-ESM2G | 144 | 90 | Yes | Yes |
| GFDL-ESM2M | 144 | 90 | Yes | Yes |
| HadCM3 | 96 | 73 | Yes | Yes |
| HadGEM2-AO | 192 | 145 | Yes | Yes |
| HadGEM2-CC | 192 | 145 | Yes | Yes |
| HadGEM2-ES | 192 | 145 | Yes | Yes |
| INMCM4 | 180 | 120 | Yes | Yes |
| IPSL-CM5A-LR | 96 | 96 | Yes | Yes |
| IPSL-CM5A-MR | 144 | 143 | Yes | Yes |
| IPSL-CM5B-LR | 96 | 96 | Yes | Yes |
| MIROC-ESM | 128 | 64 | Yes | Yes |
| MIROC-ESM-CHEM | 128 | 64 | Yes | Yes |
| MIROC4h | 640 | 320 | Yes | Yes |
| MIROC5 | 256 | 128 | Yes | Yes |
| MPI-ESM-LR | 192 | 96 | Yes | Yes |
| MPI-ESM-MR | 192 | 96 | Yes | Yes |
| MPI-ESM-P | 192 | 96 | Yes | Yes |
| MRI-CGCM3 | 320 | 160 | Yes | Yes |
| MRI-ESM1 | 320 | 160 | Yes | Yes |
| NorESM1-M | 144 | 96 | Yes | Yes |

| CMIP6 Models | Longs | Lats | TAS | PR |
|---|---|---|---|---|
| ACCESS-CM2 | 192 | 144 | Yes | Yes |
| ACCESS-ESM1-5 | 192 | 145 | Yes | Yes |
| AWI-CM-1-1-MR | 384 | 192 | Yes | Yes |
| AWI-ESM-1-1-LR | 192 | 96 | Yes | Yes |
| BCC-ESM1 | 128 | 64 | Yes | Yes |
| CESM2 | 288 | 192 | Yes | Yes |
| CESM2-FV2 | 144 | 96 | Yes | Yes |
| CESM2-WACCM | 288 | 192 | Yes | Yes |
| CESM2-WACCM-FV2 | 144 | 96 | Yes | Yes |
| CMCC-CM2-HR4 | 288 | 192 | Yes | Yes |
| CMCC-CM2-SR5 | 288 | 192 | Yes | Yes |
| CMCC-ESM2 | 288 | 192 | Yes | Yes |
| CanESM5 | 128 | 64 | Yes | Yes |
| E3SM-1-0 | 360 | 180 | Yes | Yes |
| E3SM-2-0 | 360 | 180 | Yes | Yes |
| E3SM-2-0-NARRM | 360 | 180 | Yes | Yes |
| EC-Earth3 | 512 | 256 | Yes | Yes |
| EC-Earth3-AerChem | 512 | 256 | Yes | Yes |
| EC-Earth3-CC | 512 | 256 | Yes | Yes |
| EC-Earth3-Veg | 512 | 256 | Yes | Yes |
| EC-Earth3-Veg-LR | 320 | 160 | Yes | Yes |
| FGOALS-f3-L | 288 | 180 | Yes | Yes |
| FGOALS-g3 | 180 | 80 | Yes | Yes |
| GFDL-CM4 | 288 | 180 | Yes | Yes |
| GFDL-ESM4 | 288 | 180 | Yes | Yes |
| GISS-E2-2-G | 144 | 90 | Yes | Yes |
| ICON-ESM-LR* | N/A | N/A | Yes | Yes |
| IITM-ESM | 192 | 94 | Yes | Yes |
| INM-CM4-8 | 180 | 120 | Yes | Yes |
| INM-CM5-0 | 180 | 120 | Yes | Yes |
| IPSL-CM5A2-INCA | 96 | 96 | Yes | Yes |
| IPSL-CM6A-LR | 144 | 143 | Yes | Yes |
| IPSL-CM6A-LR-INCA | 144 | 143 | No | Yes |
| KACE-1-0-G | 192 | 144 | Yes | Yes |
| KIOST-ESM | 192 | 96 | Yes | Yes |
| MIROC6 | 256 | 128 | Yes | Yes |
| MPI-ESM-1-2-HAM | 192 | 96 | Yes | Yes |
| MPI-ESM1-2-HR | 384 | 192 | Yes | Yes |
| MPI-ESM1-2-LR | 192 | 96 | Yes | Yes |
| MRI-ESM2-0 | 320 | 160 | Yes | Yes |
| NESM3 | 192 | 96 | Yes | Yes |
| NorCPM1 | 144 | 96 | Yes | Yes |
| NorESM2-LM | 144 | 96 | Yes | Yes |
| NorESM2-MM | 288 | 192 | Yes | Yes |
| SAM0-UNICON | 288 | 192 | Yes | Yes |
| TaiESM1 | 288 | 192 | Yes | Yes |

Table 1:  Details for each observed/reanalysis data product and CMIP model output. The columns give the model name, longitude and latitude resolution, and availability of (2m) surface temperature (TAS) and total precipitation (PR) for each dataset. All datasets were obtained on a rectangular grid except ICON-ESM-LR, which was obtained on an icosahedral grid with 10,242 total cells.

CMIP5 and CMIP6 outputs: `https://esgf-node.llnl.gov/projects/esgf-llnl/`

ERA5 hourly data on single levels from 1940 to present: `https://cds.climate.copernicus.eu/cdsapp#!/dataset/reanalysis-era5-single-levels?tab=overview`

NCEP/DOE Reanalysis II: `https://psl.noaa.gov/data/gridded/data.ncep.reanalysis2.html`

GPCP 1 Degree Daily Precipitation Estimate:  `https://www.ncei.noaa.gov/products/climate-data-records/precipitation-gpcp-daily`

# D  Full SCWD Rankings and Metric Comparison

In the following pages, we provide tables detailing the climate model rankings generated by SCWD and baseline methods from both the climate science and ML literature. First, we provide details on the implementation of the baseline metrics. See Section 4.4 for a discussion of these results.

## D.1  Baseline Climate Model Evaluation Metrics

Past metrics for climate model validation include the root mean square error (RMSE) and mean absolute error (MAE) applied to the climatologies, or long term means of the data [Gleckler et al., 2016]. To implement these approaches, we first require regridding the data to a common dimension. So, we regrid the data to a common resolution of $120 \times 60$ using the `remapnn` function from the Climate Data Operator (CDO) command line tools. This resolution was chosen to match the grid of locations over which the slices were computed for SCWD. Afterwards, we compute long term means at each location, ignoring leap days, to produce 365 values representing the climatologies for each day of the year. RMSE and MAE are computed between the climatologies for the reference dataset and each climate model at each location. The final rankings are determined by the area-weighted global mean over RMSE/MAE values at each location. Models marked as "NA" operate on a 360-day calendar, making it challenging to estimate a 365-day climatology to match the reference datasets.

## D.2  Baseline Wasserstein Distance Metrics

We compare our approach against existing Wasserstein-based metrics. Although these metrics have never been used for model validation, they can still provide a sense of what our method offers over the naive application of existing tools. We include ordinary Wasserstein distance (WD), Sliced Wasserstein Distance (SWD), and attempted to include Convolution SWD (CSWD). For these methods to work, we first have to regrid each model to a common set of grid points. However, unlike the baseline RMSE and MAE-based approaches described in the previous paragraph, none of the proposed Wasserstein methods account for area weighting of geographic data. So, we again regrid the climate models and reference datasets to a common resolution but now using an Icosahedral grid of 642 locations, which provides near-uniform spacing on the sphere to alleviate concerns about area weighting. The reduced dimensionality is not ideal for evaluating climate models, but serves to make the computation of the multivariate WD and sliced WD more feasible given the costs already incurred from regridding. Because we are using an Icosahedral grid to account for area weighting, we are unable to compute convolution sliced WD, which is designed for square grids.

CMIP6 Surface Temperature Rankings

| | SCWD (500km) | SCWD (1000km) | SCWD (2500km) | GMWD | RMSE (climatology) | MAE (climatology) | WD (icosahedral) | SWD (icosahedral) |
|---|---|---|---|---|---|---|---|---|
| NCEP Reanalysis | 1.412 | 1.197 | 0.923 | 0.308 | 1.456 | 1.341 | 63.307 | 1.455 |
| AWI–CM–1–1–MR | 1.382 | 1.277 | 1.044 | 0.308 | 1.518 | 1.309 | 86.069 | 1.214 |
| MPI–ESM1–2–HR | 1.404 | 1.294 | 1.039 | 0.389 | 1.582 | 1.368 | 85.374 | 1.322 |
| CESM2–WACCM | 1.419 | 1.311 | 1.074 | 0.277 | 1.495 | 1.288 | 85.684 | 1.276 |
| NorESM2–MM | 1.476 | 1.363 | 1.121 | 0.095 | 1.468 | 1.262 | 86.775 | 1.399 |
| CESM2 | 1.476 | 1.373 | 1.149 | 0.449 | 1.515 | 1.307 | 85.850 | 1.321 |
| GFDL–ESM4 | 1.544 | 1.443 | 1.196 | 0.377 | 1.496 | 1.263 | 87.304 | 1.289 |
| MPI–ESM1–2–LR | 1.600 | 1.450 | 1.117 | 0.070 | 1.706 | 1.483 | 89.824 | 1.539 |
| CMCC–CM2–HR4 | 1.561 | 1.459 | 1.228 | 0.722 | 1.714 | 1.506 | 89.084 | 1.409 |
| AWI–ESM–1–1–LR | 1.614 | 1.464 | 1.150 | 0.476 | 1.792 | 1.573 | 91.583 | 1.544 |
| MRI–ESM2–0 | 1.581 | 1.467 | 1.198 | 0.161 | 1.613 | 1.374 | 92.149 | 1.510 |
| CESM2–WACCM–FV2 | 1.679 | 1.526 | 1.246 | 0.452 | 1.713 | 1.489 | 91.093 | 1.641 |
| CESM2–FV2 | 1.734 | 1.584 | 1.296 | 0.331 | 1.729 | 1.501 | 92.046 | 1.658 |
| NorESM2–LM | 1.774 | 1.622 | 1.341 | 0.645 | 1.828 | 1.607 | 93.452 | 1.819 |
| ACCESS–ESM1–5 | 1.816 | 1.652 | 1.365 | 0.767 | 1.922 | 1.696 | 93.122 | 1.621 |
| TaiESM1 | 1.797 | 1.680 | 1.443 | 0.141 | 1.601 | 1.381 | 91.811 | 1.606 |
| E3SM–1–0 | 1.797 | 1.689 | 1.456 | 0.182 | 1.773 | 1.522 | 93.189 | 1.558 |
| CMCC–ESM2 | 1.802 | 1.694 | 1.455 | 0.592 | 1.645 | 1.396 | 90.681 | 1.548 |
| IPSL–CM6A–LR | 1.803 | 1.697 | 1.458 | 0.557 | 1.709 | 1.454 | 94.189 | 1.618 |
| GFDL–CM4 | 1.806 | 1.725 | 1.552 | 1.008 | 1.659 | 1.430 | 89.509 | 1.379 |
| INM–CM5–0 | 1.937 | 1.736 | 1.364 | 0.435 | 1.977 | 1.724 | 98.534 | 1.832 |
| IPSL–CM5A2–INCA | 1.895 | 1.752 | 1.450 | 0.446 | 1.992 | 1.718 | 94.499 | 1.676 |
| KACE–1–0–G | 1.960 | 1.791 | 1.499 | 0.269 | NA | NA | 96.472 | 1.835 |
| CMCC–CM2–SR5 | 1.896 | 1.791 | 1.559 | 0.702 | 1.721 | 1.463 | 91.996 | 1.677 |
| MPI–ESM–1–2–HAM | 1.949 | 1.814 | 1.521 | 0.174 | 1.897 | 1.668 | 93.734 | 1.661 |
| ACCESS–CM2 | 1.964 | 1.840 | 1.597 | 0.092 | 1.868 | 1.638 | 94.319 | 1.685 |
| EC–Earth3–Veg | 1.943 | 1.877 | 1.737 | 0.475 | 1.807 | 1.596 | 93.386 | 1.695 |
| NESM3 | 2.011 | 1.887 | 1.623 | 0.133 | 2.097 | 1.774 | 96.014 | 1.844 |
| CanESM5 | 2.104 | 1.913 | 1.567 | 0.265 | 2.123 | 1.877 | 102.338 | 2.106 |
| EC–Earth3–AerChem | 2.016 | 1.946 | 1.795 | 0.207 | 1.868 | 1.656 | 94.194 | 1.738 |
| EC–Earth3–Veg–LR | 2.051 | 1.971 | 1.817 | 0.150 | 1.993 | 1.768 | 96.038 | 1.861 |
| E3SM–2–0 | 2.072 | 1.973 | 1.775 | 0.541 | 1.999 | 1.748 | 96.480 | 1.820 |
| SAM0–UNICON | 2.091 | 1.980 | 1.747 | 0.744 | 1.718 | 1.486 | 96.725 | 1.725 |
| E3SM–2–0–NARRM | 2.087 | 1.997 | 1.815 | 0.658 | 1.990 | 1.755 | 96.134 | 1.740 |
| FGOALS–f3–L | 2.119 | 2.004 | 1.751 | 0.614 | 1.886 | 1.635 | 96.587 | 1.703 |
| INM–CM4–8 | 2.236 | 2.035 | 1.629 | 0.252 | 2.086 | 1.821 | 101.024 | 1.917 |
| EC–Earth3 | 2.156 | 2.087 | 1.939 | 0.238 | 1.955 | 1.738 | 97.007 | 1.812 |
| EC–Earth3–CC | 2.254 | 2.192 | 2.054 | 0.842 | 1.972 | 1.761 | 96.699 | 1.810 |
| KIOST–ESM | 2.430 | 2.338 | 2.139 | 0.631 | 2.178 | 1.927 | 102.481 | 2.065 |
| ICON–ESM–LR | 2.519 | 2.396 | 2.010 | 0.489 | 2.416 | 2.095 | 103.412 | 2.280 |
| NorCPM1 | 2.638 | 2.492 | 2.195 | 1.218 | 2.301 | 2.025 | 108.400 | 2.329 |
| BCC–ESM1 | 2.774 | 2.566 | 2.212 | 0.905 | 2.680 | 2.422 | 110.310 | 2.560 |
| FGOALS–g3 | 2.738 | 2.603 | 2.325 | 0.695 | 2.180 | 1.918 | 108.211 | 2.479 |
| MIROC6 | 3.006 | 2.878 | 2.611 | 1.531 | 2.360 | 2.103 | 108.782 | 2.423 |
| IITM–ESM | 2.996 | 2.905 | 2.709 | 0.692 | 2.496 | 2.258 | 110.685 | 2.525 |
| GISS–E2–2–G | 3.726 | 3.644 | 3.432 | 1.832 | 2.802 | 2.533 | 122.535 | 2.851 |

Table 2: CMIP6 model rankings for (2m) surface temperature based on similarity to the ERA5 Reanalysis. Distances are calculated using our proposed spherical convolutional WD (SCWD) as well as the global mean-based WD (GMWD), RMSE and MAE (both computed on the climatologies), WD and Sliced WD (both computed on data regridded to an icosahedral grid). For the SCWD calculations, three different range parameters are chosen for the Wendland kernel: 500km, 1000km (our proposed choice), and 2500km. Color fill is unique to each column in the table, and is calculated using ranks.

CMIP6 Total Precipitation Rankings

| | SCWD (500km) | SCWD (1000km) | SCWD (2500km) | GMWD | RMSE (climatology) | MAE (climatology) | WD (icosahedral) | SWD (icosahedral) |
|---|---|---|---|---|---|---|---|---|
| ERA5 | 1.804 | 1.133 | 0.701 | 0.252 | 1.936 | 1.349 | 155.514 | 0.928 |
| NorESM2–MM | 1.905 | 1.335 | 0.762 | 0.167 | 2.579 | 1.845 | 182.057 | 0.766 |
| EC–Earth3 | 2.028 | 1.589 | 0.968 | 0.236 | 2.608 | 1.906 | 175.419 | 1.136 |
| CESM2 | 2.425 | 1.597 | 0.891 | 0.263 | 2.678 | 1.930 | 184.416 | 0.945 |
| EC–Earth3–CC | 2.043 | 1.613 | 0.998 | 0.275 | 2.660 | 1.945 | 177.141 | 1.159 |
| CESM2–WACCM | 2.399 | 1.640 | 0.889 | 0.246 | 2.648 | 1.909 | 182.414 | 0.903 |
| EC–Earth3–Veg | 2.083 | 1.646 | 1.009 | 0.249 | 2.656 | 1.936 | 176.401 | 1.183 |
| EC–Earth3–Veg–LR | 2.080 | 1.647 | 1.006 | 0.193 | 2.618 | 1.902 | 172.882 | 1.264 |
| EC–Earth3–AerChem | 2.120 | 1.668 | 1.031 | 0.241 | 2.656 | 1.939 | 176.499 | 1.128 |
| MIROC6 | 2.592 | 1.683 | 0.946 | 0.486 | 2.840 | 2.048 | 195.362 | 1.294 |
| NorESM2–LM | 2.137 | 1.686 | 0.972 | 0.182 | 2.674 | 1.916 | 180.257 | 1.051 |
| CESM2–FV2 | 2.134 | 1.690 | 1.014 | 0.265 | 2.648 | 1.910 | 176.164 | 1.129 |
| TaiESM1 | 2.593 | 1.723 | 1.028 | 0.384 | 2.652 | 1.948 | 180.010 | 0.995 |
| MRI–ESM2–0 | 2.453 | 1.749 | 1.044 | 0.304 | 2.767 | 2.014 | 186.046 | 1.148 |
| CESM2–WACCM–FV2 | 2.199 | 1.750 | 1.037 | 0.262 | 2.696 | 1.948 | 176.087 | 1.249 |
| KACE–1–0–G | 2.789 | 1.759 | 1.044 | 0.340 | NA | NA | 203.446 | 1.868 |
| MPI–ESM1–2–HR | 2.288 | 1.760 | 1.040 | 0.215 | 2.765 | 1.994 | 183.813 | 1.140 |
| CMCC–ESM2 | 2.602 | 1.787 | 1.073 | 0.402 | 2.675 | 1.969 | 179.970 | 1.058 |
| ACCESS–CM2 | 2.816 | 1.796 | 1.062 | 0.447 | 3.059 | 2.179 | 209.684 | 2.051 |
| E3SM–1–0 | 2.432 | 1.800 | 1.055 | 0.359 | 2.715 | 1.976 | 178.568 | 1.154 |
| AWI–CM–1–1–MR | 2.327 | 1.802 | 1.074 | 0.234 | 2.769 | 2.009 | 185.090 | 1.288 |
| CMCC–CM2–SR5 | 2.578 | 1.818 | 1.115 | 0.412 | 2.710 | 1.986 | 179.817 | 1.104 |
| CMCC–CM2–HR4 | 2.601 | 1.818 | 1.112 | 0.291 | 2.827 | 2.015 | 186.971 | 1.103 |
| GFDL–CM4 | 3.064 | 1.836 | 0.917 | 0.225 | 2.832 | 1.980 | 199.471 | 1.425 |
| MPI–ESM1–2–LR | 2.353 | 1.851 | 1.093 | 0.167 | 2.726 | 1.973 | 177.289 | 1.380 |
| E3SM–2–0 | 3.042 | 1.861 | 0.967 | 0.276 | 2.736 | 1.942 | 184.789 | 1.033 |
| E3SM–2–0–NARRM | 3.120 | 1.873 | 0.953 | 0.276 | 2.732 | 1.943 | 187.589 | 1.030 |
| IPSL–CM6A–LR | 2.845 | 1.895 | 1.143 | 0.345 | 2.948 | 2.098 | 203.981 | 1.700 |
| IPSL–CM6A–LR–INCA | 2.926 | 1.952 | 1.163 | 0.349 | 2.963 | 2.101 | 204.900 | 1.800 |
| IITM–ESM | 2.616 | 1.964 | 1.177 | 0.240 | 2.937 | 2.115 | 188.449 | 1.209 |
| NESM3 | 2.446 | 1.967 | 1.196 | 0.181 | 2.937 | 2.073 | 184.775 | 1.083 |
| GFDL–ESM4 | 3.194 | 1.970 | 1.008 | 0.271 | 2.932 | 2.040 | 202.174 | 1.601 |
| INM–CM5–0 | 2.973 | 1.971 | 1.207 | 0.359 | 2.844 | 2.069 | 190.630 | 1.230 |
| ACCESS–ESM1–5 | 2.827 | 1.979 | 1.158 | 0.546 | 2.965 | 2.152 | 197.594 | 1.538 |
| NorCPM1 | 2.469 | 1.980 | 1.233 | 0.080 | 2.701 | 1.935 | 172.650 | 1.530 |
| SAM0–UNICON | 3.064 | 1.984 | 1.066 | 0.332 | 2.797 | 2.015 | 187.960 | 1.094 |
| INM–CM4–8 | 2.658 | 2.027 | 1.333 | 0.368 | 2.790 | 2.061 | 180.967 | 1.307 |
| CanESM5 | 3.303 | 2.088 | 1.123 | 0.209 | 2.973 | 2.090 | 198.500 | 1.557 |
| MPI–ESM–1–2–HAM | 2.681 | 2.144 | 1.294 | 0.238 | 2.902 | 2.100 | 181.616 | 1.552 |
| FGOALS–g3 | 3.438 | 2.157 | 1.136 | 0.097 | 3.136 | 2.208 | 219.320 | 2.460 |
| KIOST–ESM | 3.008 | 2.190 | 1.239 | 0.091 | 2.757 | 1.983 | 179.854 | 1.222 |
| GISS–E2–2–G | 2.847 | 2.201 | 1.366 | 0.152 | 2.857 | 2.092 | 180.662 | 1.519 |
| AWI–ESM–1–1–LR | 2.745 | 2.214 | 1.349 | 0.160 | 2.800 | 2.035 | 176.481 | 1.739 |
| ICON–ESM–LR | 2.849 | 2.295 | 1.413 | 0.118 | 3.016 | 2.153 | 185.359 | 1.428 |
| IPSL–CM5A2–INCA | 2.996 | 2.321 | 1.380 | 0.143 | 2.796 | 2.039 | 173.032 | 1.766 |
| NCEP | 3.738 | 2.490 | 1.289 | 0.581 | 2.758 | 1.943 | 215.618 | 2.533 |
| FGOALS–f3–L | 4.614 | 2.752 | 1.264 | 0.191 | 3.155 | 2.199 | 226.689 | 2.839 |
| BCC–ESM1 | 4.575 | 3.458 | 1.491 | 0.110 | 3.141 | 2.156 | 219.888 | 2.436 |

Table 3: CMIP6 model rankings for total precipitation based on similarity to the GPCP observations. Distances are calculated using our proposed spherical convolutional WD (SCWD) as well as the global mean-based WD (GMWD), RMSE and MAE (both computed on the climatologies), WD and Sliced WD (both computed on data regridded to an icosahedral grid). For the SCWD calculations, three different range parameters are chosen for the Wendland kernel: 500km, 1000km (our proposed choice), and 2500km. Color fill is unique to each column in the table, and is calculated using ranks.

**CMIP5 Surface Temperature Rankings**

| | SCWD (500km) | SCWD (1000km) | SCWD (2500km) | GMWD | RMSE (climatology) | MAE (climatology) | WD (icosahedral) | SWD (icosahedral) |
|---|---|---|---|---|---|---|---|---|
| NCEP Reanalysis | 1.412 | 1.197 | 0.923 | 0.308 | 1.456 | 1.341 | 63.307 | 1.455 |
| MPI–ESM–P | 1.491 | 1.347 | 1.039 | 0.157 | 1.671 | 1.440 | 88.946 | 1.505 |
| MPI–ESM–MR | 1.491 | 1.356 | 1.070 | 0.292 | 1.661 | 1.431 | 89.364 | 1.506 |
| MPI–ESM–LR | 1.530 | 1.390 | 1.089 | 0.132 | 1.687 | 1.458 | 89.557 | 1.502 |
| MIROC4h | 1.620 | 1.516 | 1.292 | 0.648 | 1.638 | 1.408 | 85.834 | 1.417 |
| ACCESS1–3 | 1.807 | 1.612 | 1.238 | 0.209 | 1.826 | 1.593 | 93.936 | 1.735 |
| NorESM1–M | 1.816 | 1.634 | 1.310 | 0.485 | 1.855 | 1.623 | 94.197 | 1.775 |
| ACCESS1–0 | 1.848 | 1.683 | 1.385 | 0.055 | 1.780 | 1.531 | 94.300 | 1.633 |
| IPSL–CM5A–MR | 1.833 | 1.719 | 1.448 | 0.307 | 1.869 | 1.637 | 91.688 | 1.643 |
| GFDL–CM3 | 1.877 | 1.745 | 1.446 | 0.371 | 1.884 | 1.634 | 92.071 | 1.626 |
| HadGEM2–AO | 1.977 | 1.815 | 1.516 | 0.305 | NA | NA | 96.722 | 1.692 |
| EC–EARTH | 1.921 | 1.841 | 1.650 | 0.862 | 1.994 | 1.777 | 93.500 | 1.720 |
| HadGEM2–ES | 2.014 | 1.857 | 1.582 | 0.336 | NA | NA | 97.603 | 1.845 |
| CMCC–CMS | 1.984 | 1.865 | 1.617 | 0.157 | 1.999 | 1.767 | 97.743 | 1.813 |
| MIROC–ESM–CHEM | 2.118 | 1.870 | 1.528 | 0.322 | 2.162 | 1.902 | 98.743 | 2.045 |
| CNRM–CM5 | 2.018 | 1.891 | 1.618 | 0.303 | 1.986 | 1.720 | 97.237 | 1.828 |
| MIROC–ESM | 2.141 | 1.896 | 1.559 | 0.276 | 2.170 | 1.905 | 98.937 | 2.054 |
| GFDL–ESM2M | 2.203 | 2.049 | 1.763 | 0.192 | 2.189 | 1.909 | 100.727 | 1.912 |
| CanESM2 | 2.267 | 2.057 | 1.703 | 0.265 | 2.188 | 1.914 | 104.647 | 2.213 |
| IPSL–CM5A–LR | 2.181 | 2.059 | 1.805 | 1.111 | 2.180 | 1.936 | 97.748 | 1.977 |
| MRI–ESM1 | 2.187 | 2.084 | 1.848 | 0.108 | 2.088 | 1.808 | 98.953 | 1.946 |
| CMCC–CM | 2.197 | 2.100 | 1.861 | 0.273 | 1.981 | 1.746 | 99.122 | 1.942 |
| MRI–CGCM3 | 2.224 | 2.120 | 1.881 | 0.167 | 2.090 | 1.806 | 99.788 | 1.965 |
| GFDL–ESM2G | 2.309 | 2.155 | 1.868 | 0.348 | 2.273 | 1.987 | 101.759 | 1.975 |
| HadGEM2–CC | 2.350 | 2.205 | 1.950 | 0.628 | NA | NA | 102.304 | 2.205 |
| MIROC5 | 2.357 | 2.235 | 1.972 | 0.855 | 2.013 | 1.787 | 96.780 | 1.965 |
| CMCC–CESM | 2.614 | 2.389 | 2.088 | 0.147 | 2.549 | 2.280 | 109.000 | 2.346 |
| HadCM3 | 2.575 | 2.402 | 2.111 | 0.602 | NA | NA | 111.246 | 2.404 |
| inmcm4 | 2.623 | 2.453 | 2.101 | 0.381 | 2.358 | 2.063 | 111.153 | 2.270 |
| FGOALS–s2 | 2.877 | 2.662 | 2.253 | 1.034 | 2.558 | 2.263 | 114.429 | 2.526 |
| CSIRO–Mk3–6–0 | 2.961 | 2.763 | 2.393 | 0.977 | 2.445 | 2.127 | 115.992 | 2.612 |
| FGOALS–g2 | 3.186 | 3.000 | 2.703 | 1.680 | 2.762 | 2.492 | 121.518 | 2.864 |
| IPSL–CM5B–LR | 3.361 | 3.238 | 2.929 | 0.626 | 2.741 | 2.421 | 117.348 | 2.731 |

Table 4: CMIP5 model rankings for (2m) surface temperature based on similarity to the ERA5 Reanalysis. Distances are calculated using our proposed spherical convolutional WD (SCWD) as well as the global mean-based WD (GMWD), RMSE and MAE (both computed on the climatologies), WD and Sliced WD (both computed on data regridded to an icosahedral grid). For the SCWD calculations, three different range parameters are chosen for the Wendland kernel: 500km, 1000km (our proposed choice), and 2500km. Color fill is unique to each column in the table, and is calculated using ranks.

# CMIP5 Total Precipitation Rankings

| | SCWD (500km) | SCWD (1000km) | SCWD (2500km) | GMWD | RMSE (climatology) | MAE (climatology) | WD (icosahedral) | SWD (icosahedral) |
|---|---|---|---|---|---|---|---|---|
| ERA5 | 1.804 | 1.133 | 0.701 | 0.252 | 1.936 | 1.349 | 155.514 | 0.928 |
| ACCESS1–0 | 2.257 | 1.665 | 1.033 | 0.410 | 2.887 | 2.099 | 196.045 | 1.417 |
| HadGEM2–ES | 2.274 | 1.668 | 1.039 | 0.394 | NA | NA | 196.479 | 1.339 |
| HadGEM2–CC | 2.337 | 1.713 | 1.058 | 0.364 | NA | NA | 195.209 | 1.274 |
| HadGEM2–AO | 2.387 | 1.733 | 1.067 | 0.428 | NA | NA | 199.118 | 1.446 |
| MIROC5 | 2.380 | 1.752 | 1.082 | 0.533 | 2.772 | 2.038 | 185.439 | 1.132 |
| MIROC4h | 2.809 | 1.817 | 1.080 | 0.278 | 2.937 | 2.053 | 204.359 | 1.765 |
| EC–EARTH | 2.334 | 1.859 | 1.156 | 0.199 | 2.581 | 1.879 | 171.102 | 1.462 |
| ACCESS1–3 | 2.642 | 1.913 | 1.146 | 0.487 | 2.930 | 2.138 | 193.608 | 1.500 |
| FGOALS–g2 | 2.774 | 1.913 | 1.109 | 0.132 | 2.969 | 2.104 | 202.659 | 1.566 |
| NorESM1–M | 2.435 | 1.937 | 1.212 | 0.147 | 2.725 | 1.959 | 172.984 | 1.463 |
| CanESM2 | 2.767 | 1.989 | 1.185 | 0.107 | 2.719 | 1.961 | 179.564 | 1.251 |
| CMCC–CMS | 2.498 | 2.002 | 1.214 | 0.251 | 2.893 | 2.080 | 192.831 | 1.194 |
| MPI–ESM–LR | 2.527 | 2.009 | 1.169 | 0.275 | 2.896 | 2.091 | 190.268 | 1.292 |
| GFDL–CM3 | 2.695 | 2.011 | 1.194 | 0.319 | 2.771 | 2.006 | 177.580 | 1.441 |
| HadCM3 | 2.550 | 2.027 | 1.266 | 0.234 | NA | NA | 176.560 | 1.744 |
| CanCM4 | 2.855 | 2.028 | 1.188 | 0.111 | 2.713 | 1.954 | 179.501 | 1.257 |
| CMCC–CM | 3.011 | 2.049 | 1.156 | 0.227 | 3.057 | 2.173 | 206.521 | 1.803 |
| CMCC–CESM | 2.685 | 2.129 | 1.319 | 0.204 | 2.798 | 2.045 | 176.287 | 1.653 |
| MPI–ESM–MR | 2.652 | 2.129 | 1.253 | 0.331 | 2.956 | 2.127 | 193.569 | 1.328 |
| MPI–ESM–P | 2.686 | 2.148 | 1.255 | 0.265 | 2.927 | 2.119 | 190.158 | 1.335 |
| IPSL–CM5A–MR | 2.914 | 2.189 | 1.315 | 0.144 | 2.787 | 2.030 | 177.914 | 1.576 |
| MIROC–ESM | 2.727 | 2.211 | 1.399 | 0.134 | 2.718 | 1.965 | 170.350 | 1.896 |
| MIROC–ESM–CHEM | 2.749 | 2.230 | 1.417 | 0.118 | 2.715 | 1.961 | 170.061 | 1.907 |
| IPSL–CM5A–LR | 2.882 | 2.230 | 1.333 | 0.080 | 2.709 | 1.968 | 169.559 | 1.867 |
| GFDL–ESM2G | 2.971 | 2.242 | 1.353 | 0.309 | 2.765 | 2.009 | 176.612 | 1.568 |
| MRI–CGCM3 | 3.590 | 2.276 | 1.248 | 0.237 | 3.148 | 2.204 | 214.016 | 2.192 |
| GFDL–ESM2M | 3.280 | 2.279 | 1.232 | 0.317 | 2.772 | 2.005 | 178.057 | 1.386 |
| MRI–ESM1 | 3.698 | 2.288 | 1.243 | 0.257 | 3.164 | 2.210 | 216.396 | 2.297 |
| CSIRO–Mk3–6–0 | 2.898 | 2.326 | 1.404 | 0.213 | 2.913 | 2.136 | 182.964 | 1.520 |
| FGOALS–s2 | 3.199 | 2.367 | 1.342 | 0.086 | 2.946 | 2.109 | 194.122 | 1.432 |
| CNRM–CM5 | 3.764 | 2.412 | 1.218 | 0.384 | 2.828 | 2.039 | 189.153 | 1.254 |
| NCEP | 3.738 | 2.490 | 1.289 | 0.581 | 2.758 | 1.943 | 215.618 | 2.533 |
| inmcm4 | 3.151 | 2.570 | 1.621 | 0.481 | 2.689 | 2.069 | 168.184 | 2.351 |
| IPSL–CM5B–LR | 4.504 | 2.630 | 1.168 | 0.145 | 3.228 | 2.196 | 219.657 | 2.561 |

Table 5: CMIP5 model rankings for total precipitation based on similarity to the GPCP observations. Distances are calculated using our proposed spherical convolutional WD (SCWD) as well as the global mean-based WD (GMWD), RMSE and MAE (both computed on the climatologies), WD and Sliced WD (both computed on data regridded to an icosahedral grid). For the SCWD calculations, three different range parameters are chosen for the Wendland kernel: 500km, 1000km (our proposed choice), and 2500km. Color fill is unique to each column in the table, and is calculated using ranks.

# E    Investigation of Differences Between Rankings

The impacts of climate change on temperature are not limited to changes in the local means and may include changes in variance or other moments. This phenomenon is documented in Hansen et al. [2012], where extreme tail behavior in local temperature anomalies are shown to occur alongside warming in the mean climate state. We construct the following synthetic experiment to assess the ability of each climate model evaluation method (SCWD, GMWD, RMSE, and MAE) to detect changes in regional means and variances simultaneously. Based on these findings, we re-examine the CMIP6 surface temperature rankings in Table 2 and show that RSME/MAE rank SAM0-UNICON artificially high due to their inability to detect variance changes in the temperature anomalies.

## E.1    Synthetic Experiment

To generate a realistic synthetic example, we start with the ERA5 data and linearly transform it to modify its mean and variance structure. Specifically, let $Y$ be the ERA5 reanalysis surface temperature data. $Y$ can be separated into the climatology, $C$, which is the temporal mean climate state for each location, and the anomalies $A = Y - C$. We compare $Y = C + A$ to modified datasets of the form:

$$Y_{M,s} = C + M + s * A$$

which are alternate versions of ERA5 with a mean shift of $M$ and anomaly scale factor of $s$. We consider $s = 1.1$, $s = 1.3$, and $s = 1.5$, which respectively represent 10%, 30%, and 50% increases in the variance of the anomalies. We also consider three cases of $M$ which vary in the Northern and Southern Hemispheres (NH and SH), including $(+0.5K, +0K)$ in NH/SH, $(+1K, -0.5K)$ in NH/SH, and $(+1.5K, -1K)$ in NH/SH. These are chosen to provide the same global mean change with different interhemispheric temperature asymmetries, which are an important climate feature [Friedman et al., 2013]. For each value of $M$ and $s$, we compare $Y_{M,s}$ to $Y$ using all four metrics.

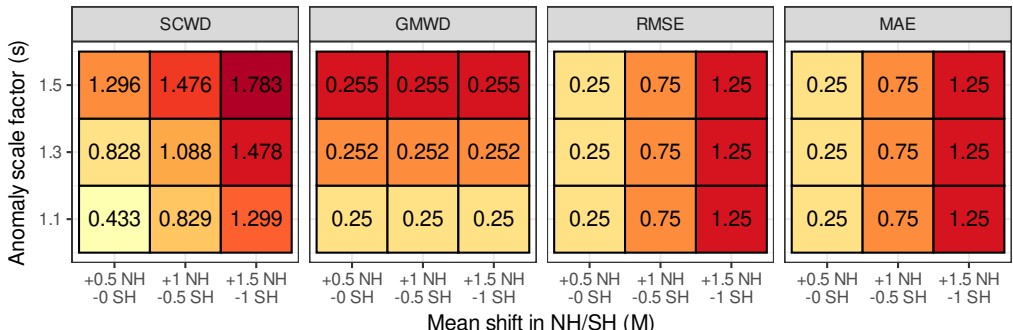

Figure 5: Results from the simulation comparing the original ERA5 data to synthetic modifications. Each panel gives the results for a different distance function. The y axis represents the anomaly scale parameter, $s$, and the x axis represents the $M$ parameter which controls mean shift in the northern/southern hemispheres. For each method, the distance is provided from the original ERA5 data to the modified ERA5 data with each of the nine combinations of $M$ and $s$. The color fill is determined by the rankings within each method, with light yellow representing a low ranking and dark red representing a high ranking.

The results are shown in Figure 5. Because each $M$ has the same global mean, GMWD does not change with $M$, although it does (very slightly) increase for higher values of $s$. Conversely, $s$ scales only the variance of the anomalies and does not impact the climatology, so RMSE/MAE are only able to detect changes in $M$, not $s$. Only SCWD detects changes in both the local means and the variance of the anomalies in this example.

## E.2    Investigating Differences in CMIP6 Rankings

The experiment demonstrates that SCWD is able to distinguish additional sources of variability beyond just differences in the climatological mean state. To understand how this ability impacts our climate model rankings, we investigate SAM0-UNICON, which ranks poorly using SCWD relative

to RMSE/MAE. Looking at the SCWD map in the supplemental material, SAM0-UNICON exhibits the highest local WD values in regions such as the coast of Antarctica. We average over the region where the local WD is the highest and compare the climatologies and anomalies for ERA5 and SAM0-UNICON in Figure 6.

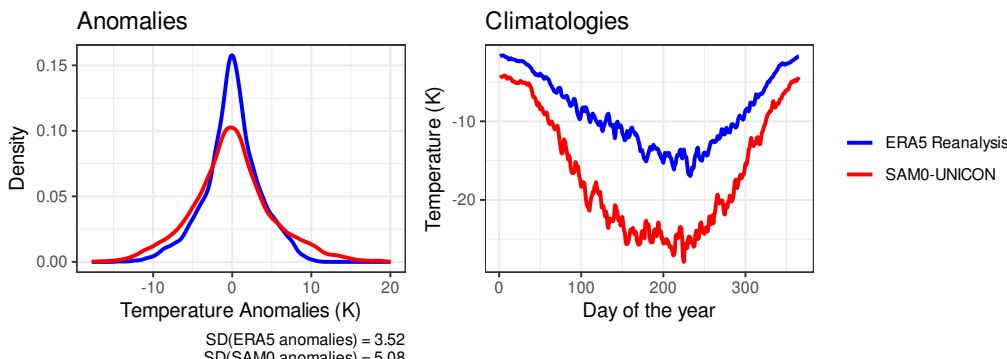

Figure 6: Comparison of anomalies and climatologies in the region where SAM0-UNICON has the largest local WD values (greater than 7.91). ERA5 is shown in blue and SAM0-UNICON is shown in red. The left plot is a density plot showing the distribution of the anomalies, and the right plot is a time series plot showing the climatologies starting on January 1st and ending on December 31st.

Relative to ERA5, we see a large difference in the climatology for SAM0-UNICON, which will impact both SCWD and RMSE/MAE. However, there is also a large increase in the variance of the anomalies in SAM0-UNICON, which is only measured by SCWD. We have demonstrated that this region has both high local WD values and large differences that are undetected by typical comparisons of the climatologies. So, this region is likely responsible in part for the harsher rankings of SAM0-UNICON by SCWD compared to RMSE/MAE.

