# OpenReview forum: "Validating Climate Models with Spherical Convolutional Wasserstein Distance"
_NeurIPS.cc/2024/Conference — NeurIPS 2024 spotlight_

### Official Review · Reviewer_vRQU · 2024-07-11

**Soundness:** 3
**Presentation:** 3
**Contribution:** 1
**Rating:** 5
**Confidence:** 4

**Summary:**

The paper proposes a new distance measure based on Wasserstein distance for data on a sphere. The work applies the methodology to climate model data, with primary focus on ranking climate models based on their agreement with reanalysis data.

**Strengths:**

The paper is well-written and easy to follow. It provides adequate background discussion on both, the methodology and the specific problem of climate model inter-comparison. The methodology is introduced rigorously, carefully defining the terms and the associated spaces. The experiment section includes a large number of models (which is not a small undertaking given the size of climate simulation data).

**Weaknesses:**

While the specific methodology is novel and may be of interest beyond climate modelling, it is a minor extension to the existing methods. Furthermore, the paper is heavily focused on a specific application. Consequently, a venue that is primarily focused on climate informatics would be more appropriate.

**Questions:**

From what I understand, the data used in daily averages of temperature and precipitation for a historic period. Is the distance calculated on the daily basis? If so, how do you take into account the fact that climate models are known to be poorly temporally aligned with observation data? Would it make more sense to aggregate the data temporally? If so, do you have any thoughts on optimal ways to do that (i.e. how to pick the optimal window for aggregation)?

You mention that baseline methods (e.g. RMSE) are unable to detect the variance of the anomalies. Do you have any thoughts on how your proposed methodology performs in the tails of the distributions that you're comparing? In most situations, the only parts of the climate distributions that are of interest to the end users are the extremes.

Do you have any thoughts on whether it makes sense to rank climate models in the first place? I assume the ranking can then be used to create a weighted ensemble of models.

Minor:
Line 52: for such a purpose
Fig. 2 y-axis  - preCipitation

**Limitations:**

Both, methodological and application specific limitations are discussed in Sec. 5.

---

> ### Author Rebuttal · Authors · 2024-08-06
>
> Thank you for taking the time to provide thoughtful feedback, we appreciate your attention to detail.
>
> First, in response to your “Contribution” score of 1 and comment:
> >**While the specific methodology is novel and may be of interest beyond climate modelling, it is a minor extension to the existing methods. Furthermore, the paper is heavily focused on a specific application. Consequently, a venue that is primarily focused on climate informatics would be more appropriate.**
>
> Please see our above rebuttal to all reviewers for our response. We believe our work (both in methods and applications) provides significant contributions to ML and climate science. We hope the above discussion better demonstrates our work's impact and potential in addressing climate and traditional ML applications.
>
> In response to your remaining comments:
> >**From what I understand, the data used in daily averages of temperature and precipitation for a historic period. Is the distance calculated on the daily basis? If so, how do you take into account the fact that climate models are known to be poorly temporally aligned with observation data?**
>
> We would like to clarify the misunderstanding that the distance was calculated on a daily basis. As you mentioned, climate models are known to be poorly temporally aligned with observation data, so no method (including ours) calculates the distance on a daily basis. Many methods focus on temporally aggregated data, such as seasonal/annual means, but fail to assess extremes that happen on shorter time scales. In contrast, our method compares the *distributions* of daily model output to the observations. Our use of unaggregated daily data allows us to reasonably approximate the distribution generated by the models. The optimal transport approach matches similar observations from model outputs and observations to avoid the time alignment issue.
> >**Would it make more sense to aggregate the data temporally? If so, do you have any thoughts on optimal ways to do that (i.e. how to pick the optimal window for aggregation)?**
>
> Most conventional methods focus on temporally aggregated data, such as annual means or long-term climatologies, to avoid the issues you mentioned above. However, the impacts of climate change on, e.g., temperature are not limited to changes in the means and may include changes in variance or other moments. Changes in the distribution tails (extremes) can have high impacts on society, thus the distribution of daily data is important to climate model evaluation and provides vital information for climate change mitigation. For example, understanding extremes for daily precipitation would help us to understand the potential for floods/droughts in different regions, which is important for agriculture and water management. Model representation of daily temperature is closely related to the model’s capability in projecting heatwaves, an important issue for public health.
> >**You mention that baseline methods (e.g. RMSE) are unable to detect the variance of the anomalies. Do you have any thoughts on how your proposed methodology performs in the tails of the distributions that you're comparing? In most situations, the only parts of the climate distributions that are of interest to the end users are the extremes.**
>
> In Appendix E.1, we provide an example where RMSE is not able to detect changes in the variance of the anomalies, but SCWD is. This is because increasing the variance creates compensating high and low extremes that cancel out when computing temporal averages. One of the strengths of SCWD is that the underlying Wasserstein distance considers all moments and quantiles of the distributions, including extremes. We demonstrate this for a specific climate model in Appendix E.2. The results suggest that differences in the rankings from SCWD and RMSE come from the fact that SCWD accounts for differences in the tail behavior of two distributions.
>
> Climate models can be evaluated from different perspectives. We focus on the entire distribution of the synthetic climate as all characteristics including mean, variance, and extremes are of interest to climate modelers. Of course, extremes are more connected to events such as natural disasters, which are of keen interest to end users. To focus only on the extreme climate, we can adapt our method to consider only certain quantiles of interest. This is similar to the trimmed Wasserstein distance discussed in [7], but we would focus solely on the tails. Our theoretical properties would be maintained in this case.
> >**Do you have any thoughts on whether it makes sense to rank climate models in the first place? I assume the ranking can then be used to create a weighted ensemble of models.**
>
> Quantitative evaluation of climate models often yields a ranking, i.e. [8] [9] [10]. Such rankings are dependent on the evaluation method and the variables being evaluated and need to be interpreted in a proper context. However, since quantitative evaluations provide users with important information about model performance, rankings can be used to create weighted multi-model ensembles as you pointed out or be used to curate a subset of high-performing models for specific applications. Model evaluation via quantitative metrics is essential for tuning climate model parameters, which was traditionally done manually. Beyond the rankings, we also showed our method's utility in producing maps to understand where models differ from historical data.
>
> Lastly, we thank you for catching those two typos, they have been fixed for the latest version of the paper!
>
> [7] Manole, T et al. Minimax confidence intervals for the sliced Wasserstein distance. 2022
>
> [8] Gleckler, PJ et al. Performance metrics for climate models. 2008
>
> [9] Schaller, N et al. Analyzing precipitation projections: A comparison of different approaches to climate model evaluation. 2011
>
> [10] Vissio, G et al. Evaluating the performance of climate models based on Wasserstein distance. 2020

---

> > ### Comment · Reviewer_vRQU · 2024-08-10
> >
> > Thank you for the thorough response. I particularly appreciate the comments on suitability for the venue, and I will increase my original score. However, I still feel that the paper would have a greater impact in a more climate-focused venue.

---

### Official Review · Reviewer_qRTk · 2024-07-12

**Soundness:** 3
**Presentation:** 3
**Contribution:** 3
**Rating:** 7
**Confidence:** 4

**Summary:**

The paper defines SCWD as a special case of their proposal for functional sliced WD which it uses to compare CMIP members against reanalysis data. Additionally, with this new distance, it analyses the effectiveness of CMIP phase 6 over phase 5

**Strengths:**

- The paper presents its ideas succinctly
- Motivates the need to find a good distance measure for comparing distributions of functions defined over $L^2(S^2)$
- Provides a smooth transition from sliced WD to its functional variant
- Method seems robust to kernel parameter
- Allows easy visualization of the differences and helps isolate regions where the fields diverge
- Validation experiments were sufficiently extensive

**Weaknesses:**

- Would have been interesting to see VAEs as a baseline as proposed by [1]

[1] Mooers, G., Pritchard, M., Beucler, T., Srivastava, P., Mangipudi, H., Peng, L., ... & Mandt, S. (2023). Comparing storm resolving models and climates via unsupervised machine learning. Scientific Reports, 13(1), 22365.

**Questions:**

- L258 suggests that the resolutions between multiple outputs isn't consistent throughout? Have they been readjusted to similar spatial dimensions?

**Limitations:**

Yes

---

> ### Author Rebuttal · Authors · 2024-08-06
>
> Thank you for your feedback, we particularly appreciate your attention to our climate application!
>
> In response to your comment:
>
> >**Would have been interesting to see VAEs as a baseline as proposed by** (Mooers et al.)
>
> We’ve added this paper to our literature review in the introduction. We believe this could make for an interesting future comparison, as both methods can be used to compare climate models and learn patterns of spatial variability. In our analysis, we focus on a single layer of convolutions and our results highlight regions where the local climate distribution of each model disagrees with observations. On the other hand, the VAE method identifies broader spatial patterns of variability that are shared between clusters of models. Both types of differences are interesting to climate scientists. For our application, we are curious about the computational speed of the VAE method. One of the advantages of SCWD is the computational speed - for example, for a single range parameter value, SCWD calculations were performed for all of our models of interest in less than 24 hours on a personal computer (the full analysis took longer as we considered multiple range values and other metrics). We are also curious about the amount of training data required to train the VAE. We do not have time to add this to our paper, but we are also very interested in this comparison and will investigate further.
>
> In regards to your comment:
>
> >**L258 suggests that the resolutions between multiple outputs isn't consistent throughout? Have they been readjusted to similar spatial dimensions?**
>
> Indeed, each model is obtained at a different spatial resolution. One of the challenges with implementing our method is that the underlying functional convolutions are continuous while the data are discrete. In our code implementation (detailed in Appendix B), we handle this by first performing a (computationally cheap) one nearest neighbor (1NN) regridding to a high resolution grid. Typical climate model validation analyses use more sophisticated (and expensive) regridding techniques to handle spatial interpolation, however, interpolation in SCWD is handled as a natural part of the metric via the kernel convolution slicing. So, the only reason we apply 1NN upsampling is to enable a high-resolution approximation of the underlying continuous kernel. The functional kernel represents a circular region on the Earth’s surface, so the high-resolution grid allows, for example, for partial weighting of the (rectangular) grid cells on the edge of the kernel radius. Intuitively, our convolution method is able to "slice" circular regions out of the original rectangular model grids.
>
> Alternatively, an exact approach with no regridding is possible by taking the integral of the kernel function over each grid cell to obtain weights for each pixel in the model output grid. The slices would be obtained by multiplying those weights by the temperature/precipitation values associated with each pixel and taking the sum. However, this would be much more computationally expensive because it would require spatial geometry operations to be performed for each different model output grid. Additionally, even though the results from the slicing process would be exact, the analysis would still be limited by the available spatial resolution of the model outputs!

---

> > ### Comment · Reviewer_qRTk · 2024-08-11
> >
> > Thank you for the response and for clarifying how you handled regridding. I find this paper particularly valuable to climate science community, and I will increase my original score.

---

### Official Review · Reviewer_UJm5 · 2024-07-12

**Soundness:** 4
**Presentation:** 3
**Contribution:** 4
**Rating:** 7
**Confidence:** 5

**Summary:**

The paper introduces a new method for validating climate models by comparing their outputs to reanalysis data. The proposed method, Spherical Convolutional Wasserstein Distance (SCWD), accounts for spatial variability and local differences in the distribution of climate variables. The authors apply SCWD to evaluate historical model outputs from the Coupled Model Intercomparison Project (CMIP) phases 5 and 6, demonstrating modest improvements in the phase 6 models in producing realistic climatologies. The technical claims are well-supported by thorough theoretical and empirical analyses. The authors provide a robust mathematical foundation for SCWD and demonstrate its effectiveness in capturing spatial variability through extensive experiments. The paper is well-structured and written, with detailed explanations of the methodology and comprehensive evaluation results. However, some sections could benefit from additional clarity, particularly the mathematical derivations and kernel selection process.

**Strengths:**

Originality: The introduction of SCWD as a new metric for climate model validation is innovative and addresses the limitations of existing methods.
Quality: The methodology is rigorously developed and supported by extensive experimental validation using real-world climate data.
Clarity: The paper provides clear explanations of the SCWD methodology, supported by visualizations and detailed examples.
Significance: The proposed method has significant implications for improving climate model validation, which is crucial for accurate climate projections and policy-making.

**Weaknesses:**

Mathematical Derivations: Some mathematical derivations, particularly those related to the convolution slicer and kernel functions, could be explained more clearly to enhance understanding.
Generalization: While the method is well-validated on historical climate data, additional experiments on different climate variables and temporal resolutions would strengthen the generalizability of the findings.

**Questions:**

Could the authors provide more details on the selection process and theoretical justification for the specific kernel function used in SCWD?
Have the authors considered applying SCWD to other climate variables or different temporal resolutions to evaluate its generalizability?

**Limitations:**

The authors adequately address the limitations of their work, including the need for device-aware optimizations and the challenges in parameter selection. They also acknowledge the potential for further improvements and generalization of SCWD, providing constructive suggestions for future research.

---

> ### Author Rebuttal · Authors · 2024-08-06
>
> Thank you for your insightful comments and suggestions. In response to your comments on including additional climate variables/temporal resolutions, i.e.
>
> >**While the method is well-validated on historical climate data, additional experiments on different climate variables and temporal resolutions would strengthen the generalizability of the findings.**
>
> We certainly hope our method will see wider use in the climate community with additional variables/resolutions. We have already seen evidence of wider interest in SCWD. For example, we were approached by a climate modeling center interested in SCWD and are working with them to apply this method to other datasets. The work is ongoing, but so far SCWD has proven useful for assessing long-term (1-10 year) daily climate forecasts and for identifying problematic regions in those forecasts. One key focus of that work is considering the performance over different time scales and different seasons.
>
> Additionally, in regard to your comments on providing additional clarifications, i.e.
>
> >**Some mathematical derivations, particularly those related to the convolution slicer and kernel functions, could be explained more clearly to enhance understanding.**
>
> We agree that some additional clarity on our theoretical details would help convey our message. We expanded the discussions of the convolution slicer and kernel selection as follows:
>
> For the **convolution slicer**,  we edited the paragraph starting line 144 as follows:
>
> "To create a valid functional sliced WD, we construct pushforward measures based on the convolution slicer $c_s(f)$. To satisfy the definition of a pushfoward measure, we must show that $c_s(f)$ is a Borel measurable function from $L^2(\mathcal S) \rightarrow \mathbb{R}$. By continuity of $k$, when location $s\in\mathcal S$ is fixed, $k(s,u)$ is a continuous function from $u\in\mathcal{S}\rightarrow\mathbb{R}$. Because $\mathcal{S}$ is compact, $k(s,u)$ is a continuous function on a compact set and is thus bounded and $L^2$-integrable. It follows that the convolution slicer $c_s(f)$ is an integral of the product of two functions $f,k \in L^2(\mathcal S)$, so by Hölder's inequality, $c_s(f)$ is a bounded linear operator from $L^2(\mathcal{S})\rightarrow\mathbb{R}$. Stein et al. [2011] states that bounded linear operators are also continuous, so $c_s(f)$ is a continuous linear operator and thus Borel measurable. So, for any measure $P\in\mathcal{P}(L^2(\mathcal{S}))$, the pushforward $c_s\\#P$ is a valid measure in $\mathcal{P}(\mathbb{R})$. Therefore, we can define a functional sliced WD between distributions in $\mathcal{P}(L^2(\mathcal{S}))$ as follows:"
>
> We hope this provides some more clarity on the formulation and properties of the convolution slicer.
>
> For the **choice of kernel function**, we considered both the Kent distribution function and Wendland function because both are positive definite on the sphere and are popular choices in spatial statistics. Of the two, we prefer the Wendland function due to its compactness, which enables sparse matrix computations in our analysis. To guarantee positive definiteness, we need the kernel to be smaller than the radius of the sphere. Beyond that, the choice is left to the user, and we chose 1,000km to balance the need for fine-scale spatial perspectives and the available spatial resolution of the data.

---

> > ### Comment · Reviewer_UJm5 · 2024-08-12
> >
> > Thank you for addressing my concerns. I will keep my positive score unchanged.

---

### Official Review · Reviewer_V9JK · 2024-07-16

**Soundness:** 4
**Presentation:** 4
**Contribution:** 3
**Rating:** 8
**Confidence:** 3

**Summary:**

Developing metrics for comparison between high dimensional, multivariate climate models is an important and open area of study . Vissio et al (2020) proposed the use of the Wasserstein distance to quantify the similarity between climate models. However this approach involves spatial averaging, and therefore significant information is lost. This work introduces functional sliced Wasserstein distance in spherical coordinates, which provides a computationally tractable Wasserstein metric without spatial averaging. The method is demonstrated by comparisons between CMIP model data and ERA reanalysis data.

**Strengths:**

The work is timely and presents a strong contribution to the field of climate science. The presentation is excellent - motivation and connections to previous work are clearly established. The new method is clearly explained, and demonstrated in a sensible set of experiments. The capability of the slicing kernel to focus on specific local regions provides a tremendous amount of flexibility to the metric, which will have utility in a wide range of important applications. Comparisons to other standard metrics are also made, and in cases where there are discrepancies with baselines, these discrepancies are discussed. Finally the authors speculate on potential applications beyond climate science.

**Weaknesses:**

The paper has no obvious weaknesses.

**Questions:**

I am wondering whether the method could be demonstrated on a simpler toy problem where the ground truth is better established, and the complexity and high dimensionality of the system is retained, before application to a reanalysis-vs-model comparison. As discussed in section 4.1, reanalsyis comparisons are still subject to discrepencies from other factors such as model physics.

**Limitations:**

Limitations of both the method (lines 107-119 and 356-359) and the results (section 4.1) are discussed.

---

> ### Author Rebuttal · Authors · 2024-08-06
>
> Thank you for your feedback and suggestions! In response to your comment:
>
> >**I am wondering whether the method could be demonstrated on a simpler toy problem where the ground truth is better established, and the complexity and high dimensionality of the system is retained, before application to a reanalysis-vs-model comparison**
>
> We shared your thoughts on this subject. While we did not have space for our full exploration in the main text, Appendix E.1 provides a synthetic data experiment that we believe addresses your concerns. We retain the high complexity of climate observations by making modifications to the mean trends and anomalies of the ERA5 data, and compare performance between metrics.
>
> In that experiment, we found that SCWD could detect changes in both the climatological means as well as the variance of the anomalies. The global mean-based Wasserstein distance failed to capture differences when the climatological mean had compensating errors in space. Baseline metrics such as RMSE/MAE, which must be computed using long-term climatological means to avoid issues with time misalignment of climate models and observations, were unable to distinguish changes in the scale of the anomalies.

---

### Author Rebuttal · Authors · 2024-08-06

Thank you all for taking your time to provide a thorough review of our work. One shared concern from a few of the reviewers was the generalizability of our method to other tasks within climate science and ML. First, we provide some further insight on our contributions by responding to a comment from **Reviewer vRQU**, then we expand the discussion section in the manuscript to highlight future opportunities for our method to address other ML tasks.

## **1. In response to the following comment from Reviewer vRQU:**

>**While the specific methodology is novel and may be of interest beyond climate modelling, it is a minor extension to the existing methods. Furthermore, the paper is heavily focused on a specific application. Consequently, a venue that is primarily focused on climate informatics would be more appropriate.**

We disagree strongly that our method is a “minor” extension to existing methods. Sliced variants of the Wasserstein distance (WD) and other optimal transport methods are a popular subject in ML. Many such works, including Sliced WD (SWD) [1], Generalized SWD [2], Convolution SWD [3], Intrinsic SWD [4] and more, are published in prestigious ML conference proceedings, especially NeurIPS. The contributions in our own paper are analogous to the extension from CNNs to Spherical CNNs [5] published in ICLR.

Furthermore, NeurIPS embraces various topics as shown in the call for papers, specifically:

> (NeurIPS 2024) is an interdisciplinary conference that brings together researchers in machine learning, ..., **natural sciences**, social sciences, and other adjacent fields. We invite submissions presenting new and original research on topics including but not limited to the following … **Applications** (e.g., vision, language, speech and audio, Creative AI) … **Machine learning for sciences (e.g. climate...)**

Although our application may be suitable for a climate informatics conference, we believe our paper as a whole is a better fit for NeurIPS. In particular, our introduction of new methodology and supporting theoretical results will be much better received by the ML community than the climate science community. Also, NeurIPS has shown a clear commitment to addressing climate issues through hosting the “Tackling Climate Change with Machine Learning” workshop series the last couple years.

Evaluation of climate models is an important topic given the impacts of climate change on society and the fact that climate models are the primary tool for climate projection. This application is a long standing and important problem in climate science, so we expect our work will be highly impactful.

## **2. Generalizability of our method and future work in ML/climate**

Furthermore, we believe our work is generalizable to tackle other important topics in climate science and more broadly in ML. In our individual responses, we have addressed the following:

+ **Reviewer UJm5:** generalizations to other climate variables and temporal resolutions. We’ve started work on these topics in collaboration with researchers from a climate modeling center who reached out to us about SCWD.
+ **Reviewer vRQU:** extension of SCWD to focus only on climate extremes, which are important to end users of climate models. A straightforward adaptation of SCWD is possible to consider certain distributional quantiles of interest.

See our responses to those reviewers for details. Additionally, we consider the following tasks as opportunities for future work using SCWD for climate science:

1. Learn an optimal value for the range parameter of our chosen Wendland kernel. The challenge would be to determine a climatologically relevant loss/criteria to evaluate different range values.
2. Similar to the neural-network based defining functions in [2], we could use estimate a neural network to use as our kernel function. Note that we chose the Wendland kernel in our current manuscript because of the importance of comparing local spatial features in climate models. However, neural network-based functions may work better for other applications.
3. Machine-assisted climate model tuning [6]. Climate models are mathematical simulations that rely on parameters to control different aspects of the model behavior. Historically, these parameters were hand-tuned, but recent work has focused on incorporating machine learning. Due to the extreme computational costs in climate modeling, quality metrics are essential for this task. This is one of the motivations for establishing the validity and utility of SCWD.

SCWD and the more general functional SWD introduced in our paper have potential for broader applications in ML. Similar to [3] which was used to train generative models for rectangular images, SCWD can be used to train generative models for $360^o$  images. Likewise, the functional SWD can be used to train generative models for functions on any manifold. Lastly, the functional SWD could be applied to texture mapping and/or color transfer (common use cases for WD variants) on the surface of 3D models, which can be considered as non-euclidean manifolds.

We’ve added much of the discussion here to Section 5 of the paper. We see our work as bridge to adapt cutting edge methods from the ML literature to address important topics in climate science and hope to make this more clear!

[1] Sliced and radon wasserstein barycenters of measures. Bonneel et al., 2015

[2] Generalized sliced Wasserstein distances. Kolouri et al., 2019 (NeurIPS)

[3] Revisiting Sliced Wasserstein on Images: From Vectorization to Convolution. Nguyen and Ho, 2022 (NeurIPS)

[4] Intrinsic Sliced Wasserstein Distances for Comparing Collections of Probability Distributions on Manifolds and Graphs. Rustamov and Majumdar, 2023 (ICML)

[5] Spherical CNNs. Cohen et al., 2018 (ICLR)

[6] Toward machine-assisted tuning avoiding the underestimation of uncertainty in climate change projections. Hourdin et al., 2023

---

### Author Response · Authors · 2024-08-07
**Visibility of global rebuttal**

Hello, we want to make sure that the reviewers are able to see our global rebuttal. On our end, it seems that they do not have reading permissions, and we were unable to add these permissions when submitting the rebuttal. We can repost that content using official comments if necessary! Thank you.

---

### Decision · Program_Chairs · 2024-09-25

**Decision:**

Accept (spotlight)

**Comment:**

The paper introduces a novel Wasserstein distance metric allowing for the comparison of climate model data, using it to compare CMIP6 models with reanalysis data, as well as the differences between CMIP6 and CMIP5. The reviewers agree that this is an excellent paper in terms of topic, results, and presentation. I accordingly strongly recommend acceptance.

A citation that may be useful to add is:

Mooers, G., Pritchard, M., Beucler, T., Srivastava, P., Mangipudi, H., Peng, L., Gentine, P. and Mandt, S., 2023. Comparing storm resolving models and climates via unsupervised machine learning. Scientific Reports, 13(1), p.22365.